# A high-resolution, nanopore-based artificial intelligence assay for DNA replication stress in human cancer cells

Mathew J. K. Jones [1,2] ✉, Subash Kumar Rai [1], Pauline L. Pfuderer [3,4], Alexis Bonfim-Melo [1], Julia K. Pagan[5], Paul R. Clarke[1,6], Francis Isidore Garcia Totañes [7], Catherine J. Merrick [3], Sarah E. McClelland [8] & Michael A. Boemo [3,9] ✉

DNA replication stress is a hallmark of cancer that is exploited by chemotherapies. Current assays for replication stress have low throughput and poor resolution whilst being unable to map the movement of replication forks genome-wide. We present a new method that uses nanopore sequencing and artificial intelligence to map forks and measure their rates of movement and stalling in melanoma and colon cancer cells treated with chemotherapies. Our method can differentiate between fork slowing and fork stalling in cells treated with hydroxyurea, as well as inhibitors of ATR, WEE1, and PARP1. These different therapies yield different characteristic signatures of replication stress. We assess the role of the intra-S-phase checkpoint on fork slowing and stalling and show that replication stress dynamically changes over S-phase. Finally, we demonstrate that this method is applicable and consistent across two different flow cell chemistries (R9.4.1 and R10.4.1) from Oxford Nanopore Technologies. This method requires sequencing on only one nanopore flow cell per sample, and the cost-effectiveness enables functional screens to determine how human cancers respond to replication-targeted therapies.

DNA replication stress is characterised by frequent slowing, stalling, and collapse of replication forks which causes unreplicated regions of the genome, DNA damage, mutations, and chromosomal instabilities that drive tumourigenesis[1,2]. Many important chemotherapeutic agents target the replication stress response (RSR) pathway[3] including inhibitors of ATR[4], PARP1[5], WEE1[6], and dNTP synthesis with hydroxyurea (HU)[7,8] or 5-fluorouracil[9], as well as DNA crosslinkers like cisplatin[10]. How these agents affect the movement and fidelity of replication forks across the genome remains unknown due to the low throughput and/or insufficient resolution of current assays. The

location of DNA breaks that result from replication fork stalls can be mapped with techniques such as BLESS[11], Break-seq[12], and TrAEL-seq[13] but these methods require the fork to stall such that it results in a break and no information about origin firing or fork movement prior to the break is captured. DNA fibre analysis is a single-molecule method that can measure fork velocity, origin density, and the frequency of fork stalling[14,15], but the throughput and spatial resolution are both low and fibres cannot be mapped to the genome without resorting to fibre-fish[16] which is not scalable to the whole genome. Recently, optical replication mapping (ORM) provided a high-throughput, single-

[1]Frazer Institute, Faculty of Health, Medicine, and Behavioural Sciences, University of Queensland, Brisbane, QLD, Australia. [2]School of Chemistry & Molecular Biosciences, University of Queensland, Brisbane, QLD, Australia. [3]Department of Pathology, University of Cambridge, Cambridge, UK. [4]Cancer Research UK Cambridge Centre, Li Ka Shing Centre, Robinson Way, Cambridge, UK. [5]School of Biomedical Sciences, The University of Queensland, Saint Lucia, QLD, Australia. [6]Institute for Biomedicine and Glycomics, Griffith University, Southport, QLD, Australia. [7]Wellcome Sanger Institute, Cambridge, UK. [8]Barts Cancer Institute, Queen Mary University of London, London, EC1M 6BQ, UK. [9]Department of Genetics, University of Cambridge, Cambridge CB2 3EH, UK. ✉e-mail: mathew.jones@uq.edu.au; mb915@cam.ac.uk

molecule approach to map origin firing genome-wide[17], but the 15-kb resolution is too low to detect fork stalling. Long-read sequencing using the Oxford Nanopore Technologies (ONT) platform has enabled the detection of replication origins and fork movement in budding yeast[18–22] as well as the malaria parasites *Plasmodium falciparum*[23,24] and *Plasmodium knowlesi*[25]. It has also been used to measure the replication rates of human mitochondrial DNA[26] as well as replication origins in the human genome[27]. Here, we introduce a method that measures fork speed and stalling on single molecules with up to single-nucleotide resolution and use it to show that chemotherapies create different "replication stress signatures". Our method can clearly distinguish between replication fork slowing and stalling, and by mapping these molecules to the genome, we assess the role of the intra-S-phase checkpoint and show that replication stress dynamically changes over S-phase.

## Results

### DNAscent detects replication forks on single molecules from human cancer cells

Our DNAscent software can detect two thymidine analogues, BrdU and EdU, on single nanopore-sequenced molecules[23]. When these base analogues are sequentially pulsed into S-phase cells, they are incorporated into the nascent strand by replication forks, leaving a "footprint" of fork movement. Therefore, we pulsed A2058 human melanoma cells and human RPE1 cells expressing PIP-FUCCI[28] with EdU, then BrdU, and followed by a thymidine chase. We then enriched for S-phase cells by fluorescence-activated cell sorting, extracted ultra-high-molecular weight DNA, and sequenced it on the Oxford Nanopore MinION platform (Fig. 1a, b; Figure S1). As shown previously in *S. cerevisiae*[18,20], fork stalling manifests as a sharp drop-off in analogue incorporation into nascent DNA (Fig. 1c) and we observed stalled forks in A2058 melanoma cells (Fig. 1d). The length (in base pairs) of the base analogue tracks, divided by the total pulse length, yielded a measure of fork speed that was reproducible between biological replicates with fork speeds of ~1.4 kb/min in RPE1 cells that are consistent with studies measuring fork speeds in RPE1 cells via DNA fibres[29–31] (Fig. 1e). We performed DNA fibre analysis for A2058 cells where we observed a difference of approximately 0.3–0.5 kb/min between fork speeds measured by DNAscent and fork speeds measured by DNA fibre (Figure S2). This difference could be due to variation in fibre stretching or differences in the resolution and sensitivity of analogue detection[32]. Our optimised ultra-long sequencing protocol yielded over 180,000 total reads with a mapping length greater than 20 kilobases (kb), an N50 read length of ~90 kb, and 4100 called forks from a single MinION flow cell (averages: 150,000 reads longer than 20 kb, 90 kb N50, 2050 fork calls; Table S1) with a false positive rate of fork calls less than 0.004% (Table S2). This read length and throughput allowed us to capture complex replication dynamics across the entire human genome with multiple forks, origins, and termination sites on the same molecule (Fig. 1f).

### DNAscent distinguishes between fork stalling and slowing under chemotherapies

We treated A2058 cells with either a PARP inhibitor (Olaparib), WEE1 inhibitor (MK1775), ATR inhibitor (VE-821), or hydroxyurea (HU) (Fig. 2a). These treatments had a marked effect on replication fork dynamics that was clearly visible on single molecules (Fig. 2b). As expected, HU[15], ATR inhibition[33], and WEE1 inhibition[34] all slowed forks (Fig. 2c). To test the method further and provide an internal control, we added HU with the BrdU pulse and verified shortening of the BrdU track but not the EdU track (Figure S3). The dose and treatment time of Olaparib mirrored that of Maya-Mendoza and colleagues[35] and, consistent with their findings using DNA fibre analysis, we observed a 30% increase in fork speed. Fork speeds from all conditions were consistent across biological replicates (Figure S4) and, for each agent, the treatment effect exceeded expected variation between replicates

(Figure S5). To create a quantitative measure of fork stalling, we defined a "stall score" to measure how abruptly BrdU incorporation ends. Stall score is a measure between 0 and 1 of the proportional decrease in the frequency of positive BrdU calls at the BrdU end of the replication fork track. Forks assigned a stall score near 0 are unlikely to be stalled and forks assigned a stall score near 1 are likely to be stalls or pauses (Fig. 3a). Stall score provides an additional layer of information about replication mobility that has the potential to distinguish between fork stalling and slowing: compared to untreated cells, HU and WEE1 inhibition resulted in fork slowing, and PARP inhibition resulted in a fork speed increase (see Fig. 2c), but all three of these chemotherapies resulted in a similar distribution of stall scores to that of untreated cells (Fig. 3b). ATR inhibition did not slow forks as much as HU, but treatment with ATR inhibitors resulted in a marked increase in stall score indicative of frequent fork stalling. While HU is generally thought to rapidly stall replication forks, multiple DNA fibre analysis studies have shown slow but continued fork progression during short and prolonged HU treatment[36,37]. Our approach builds on these DNA fibre methods by discriminating between fork slowing in cells treated with HU and rapid stalling in cells treated with ATRi. As before, stall scores from all conditions were consistent across biological replicates (Figure S6). Notably, only ATRi showed strong evidence of a treatment effect beyond the expected variation between replicates (Figure S7).

### Chemotherapies create distinct signatures of replication stress

DNA replication stress is an umbrella term that refers to frequent fork slowing and/or stalling. DNAscent can measure each of these attributes, and to unify them into an overall measure of replication stress, we represented each fork as an 8-dimensional point consisting of features pertaining to fork speed, level of analogue incorporation, and stall score (see Methods for details) and reduced points on this 8-dimensional manifold to two dimensions using Uniform Manifold Approximation and Projection (UMAP)[38]. Forks measured from cells treated with each chemotherapy clustered together, showing that disrupting different elements of the DNA replication stress response pathway creates a replication stress signature in this 8-dimensional space (Fig. 3c). To demonstrate that our new method captures fundamentally more meaningful information about DNA replication stress than existing methods that just measure fork speed, we repeated the procedure in a 5-dimensional space that excluded any measurement related to fork speed (the track lengths of EdU, BrdU, and the overall fork; see Methods). We found that stall score and the level of base analogue incorporation alone were sufficient to create distinct stress signatures of disruption to different elements of the RSR pathway (Fig. 3d). While we anticipate variation due to the timing and dosage of treatment, these results show that different chemotherapies can create a characteristic replication stress signature based on the pathway that they target.

### Mapping stressed forks to the genome reveals checkpoint-dependent changes over S-phase

To investigate whether fork stress changes across S-phase, we applied our method in HCT116 colon cancer cells to leverage existing high-resolution Repli-Seq replication timing data[39]. We assessed both wild-type HCT116 cells and HCT116 cells with a *CDK2*AF/AF mutation that prevents intra-S-phase checkpoint activation by WEE1[40]. Our measured fork speed in wild-type cells was consistent with DNA fibre analysis[41]. While we observed a higher variability between repeats than in A2058 (wild-type median fork speed of 1.7 k/min with stall score median 0.3; *CDK2*AF/AF median fork speed of 1.26 kb/min with stall score median 0.39; Figure S10) the *CDK2*AF/AF cells showed a slower fork speed and higher stall score than the wild-type within each repeat (Fig. 4a, b). Moreover, there was moderate support for a treatment effect of the *CDK2*AF/AF mutation (Figure S11). This was consistent with the effect of WEE1 inhibition on fork speed and stall score in A2058 cells, as WEE1

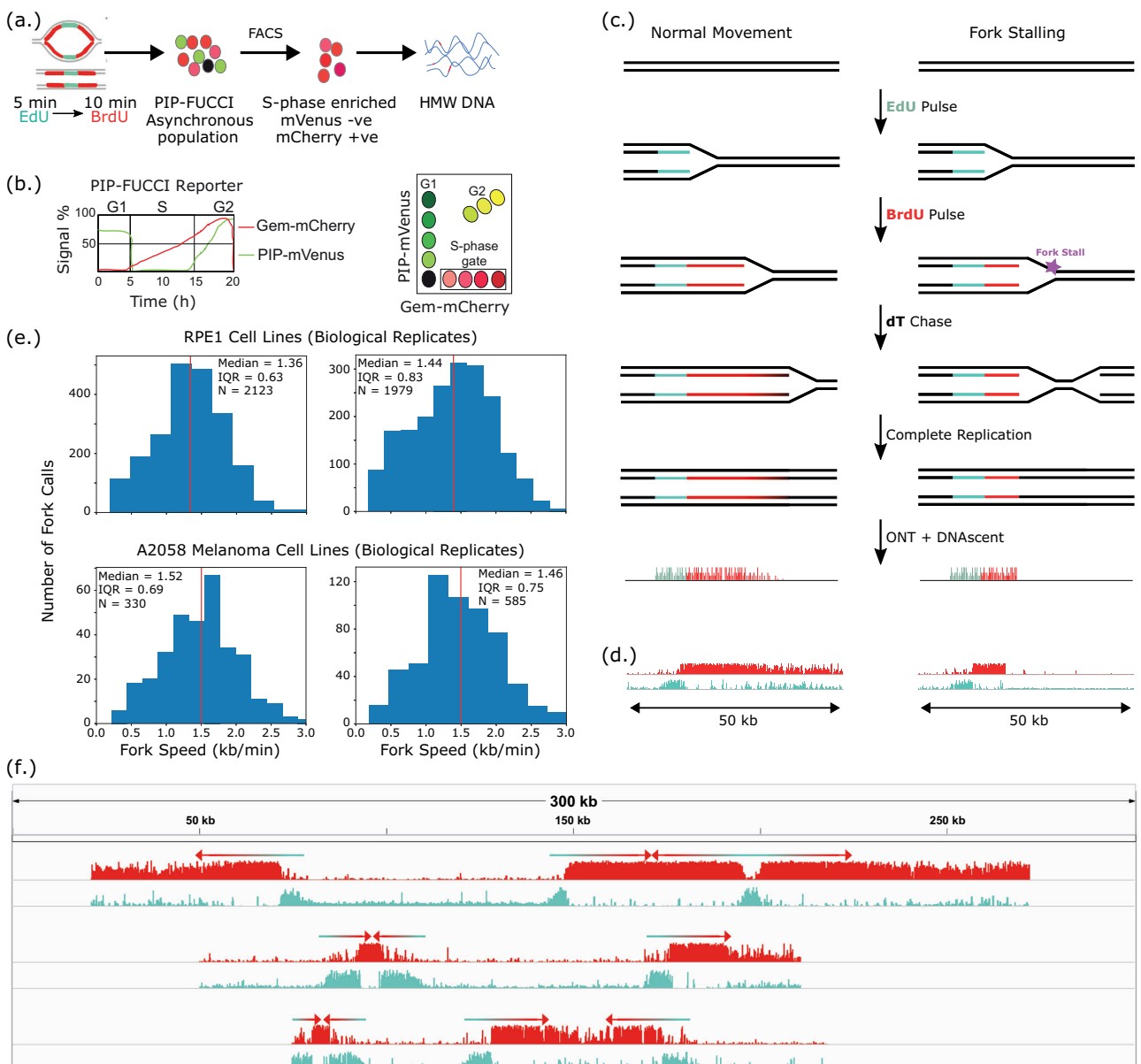

**Fig. 1 | Replication dynamics on ultra-long nanopore reads from human cancer cells. a** Cells expressing PIP-FUCCI were sequentially labelled with a 5-minute EdU pulse, a 10 min BrdU pulse, and a thymidine chase. The asynchronous population of cells was enriched for S-phase cells using FACS (mCherry+ mVenus-), and high molecular weight (HMW) DNA was extracted and sequenced on the Oxford Nanopore MinION platform. **b** PIP-FUCCI gates on high levels of geminin (Gem) and low levels of the PCNA-interacting protein (PIP) degron enabling S-phase enrichment without the cellular stress caused by arresting the cell cycle. **c** Diagram showing that fork stalling manifests as a sudden drop in BrdU incorporation into the nascent strand. **d** Two nanopore-sequenced molecules from A2058 melanoma cells analysed with DNAscent that show the patterns in (**c**). Tracks show the probability of BrdU (red) and EdU (blue) at each thymidine position along the molecule. **e** Distribution of fork speed measured by DNAscent in untreated RPE1 cells (upper) and A2058 cells (lower) for two biological replicates. Vertical red line median, IQR interquartile range, N number of fork calls. **f** Three ultra-long single molecules, each represented as a group of two tracks of bar graphs. The top track shows the DNAscent-called probability of BrdU (red) and the bottom track shows the DNAscent-called probability of EdU (blue). Each read is annotated with arrows to show fork direction. Reads were moved onto the same axis from different regions of the genome. Source data are provided as a Source Data file.

inhibition functionally mirrors a *CDK2*^AF/AF mutation (Fig. 2c; Fig. 3b). We calculated the median replication time in S-phase (Trep) for the genomic position of each called fork using high-resolution Repli-Seq replication timing data for HCT116 cells[39] in order to show how the change in fork speed and stall score over S-phase can be measured from a single sequencing run. In wild-type cells, fork speed increased (in agreement with single-cell studies[42]) and stall score decreased in genomic positions that tended to replicate later in S-phase (Fig. 4c, d). This relationship vanished in the *CDK2*^AF/AF mutant, as these cells maintained a constant slow fork speed and high stall score across regions of the genome that replicated both early and late in wild-type

cells. These trends in stall score and fork speed over S-phase for both wild-type and *CDK2*^AF/AF HCT116 cells were consistent across biological replicates (Figure S10).

## A transition to R10.4.1 nanopore chemistry
To future-proof our method and show that our results hold across different generations of Oxford Nanopore flow cell chemistries, we developed and trained DNAscent to work on Oxford Nanopore flow cells using the latest R10.4.1 flow cells. Building upon the strategies we developed to train DNAscent v2[20] and DNAscent v3[23], we developed a new version of DNAscent (v4) for the R10.4.1 chemistry. The neural network in DNAscent

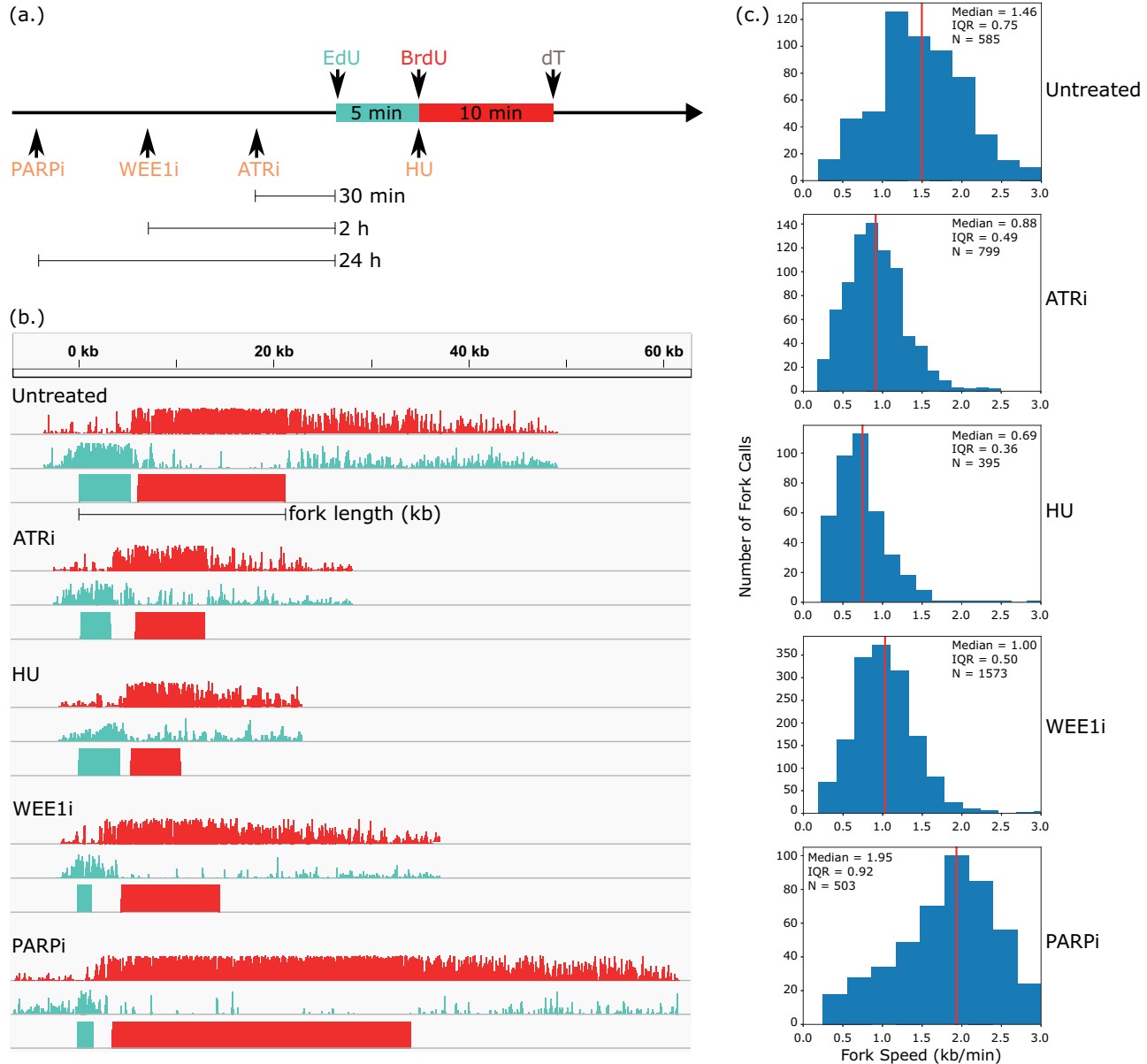

**Fig. 2 | DNAscent measures disruptions to fork speed caused by inhibiting the replication stress response pathway. a** A2058 cells were given a monotherapy of either HU, ATR inhibitor, WEE1 inhibitor, or PARP inhibitor at the specified time-points relative to the BrdU and EdU pulses. **b** Five single molecules analysed by DNAscent, each showing a rightward-moving fork from untreated cells as well as from each of the monotherapies. Forks were from different regions of the genome, moved onto the same axis so that the start of the EdU tracks align. Each fork is represented as group of three tracks showing (from top to bottom) the probability of BrdU at that position on the read, the probability of EdU at that position on the read, and DNAscent's segmentation of the read into EdU- and BrdU-positive

regions. Fork length was measured by computing the genomic distance, in kilo-bases, between far ends of the EdU and BrdU segments (shown for the top fork) and dividing this distance by the 15 min total pulse duration. **c** Distribution of fork speeds for untreated cells and cells under each monotherapy. Forks near the end of reads, as well as forks together at origins or terminations, were excluded to avoid inaccuracies in fork speed (see Methods). Vertical red line median, IQR interquartile range, N number of fork calls. Treatment effect, as measured by a hierarchical Bayesian model (see Methods), is reported in Figure S5. Source data are provided as a Source Data file.

v4 was trained using BrdU and EdU incorporated into human RPE1 cells (Figure S12; Table S3 for model architecture). We observed cleaner fork tracks with DNAscent v4, particularly a very low false positive rate following the thymidine chase (Fig. 5a). DNAscent v4 was benchmarked on human RPE1 reads as well as reads from the malaria parasite *Plasmodium falciparum* to stress test the model's performance on an outlier 81% AT-rich genome. For both human cells and *P. falciparum*, DNAscent v4 showed considerably higher positive BrdU and EdU calls for a much lower false positive rate (Figure S13; Table S4). To ensure that our observations of fork speed and stall scores were consistent across R9.4.1 and R10.4.1, we used DNAscent v4 to measure fork speed in human

RPE1 cells and found that fork speed on R10.4.1 (Fig. 5b) was consistent with fork speed on R9.4.1 as measured with DNAscent v3 (Fig. 1e). As expected, fork stalling was low. On treatment with ATRi, we again observed a decrease in fork speed and an increase in stall score when using R10.4.1 and DNAscent v4. Taken together, these data show that we can consistently estimate replication fork speed and stalling from Oxford Nanopore data across different nanopore chemistries.

## Discussion

We have developed a new method for the high-throughput, high-resolution measurement of DNA replication stress across the human

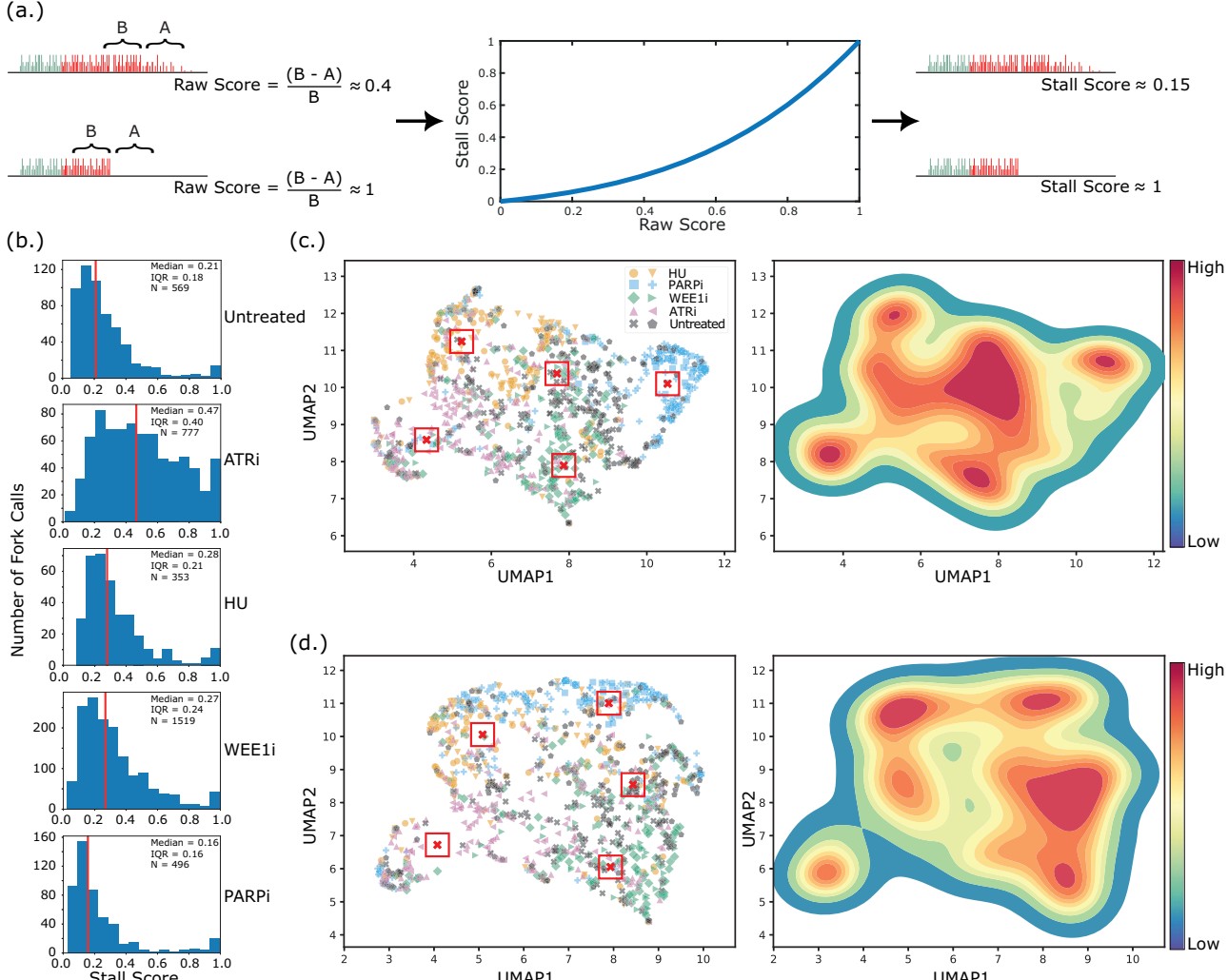

**Fig. 3 | Replication stress signatures stratify cells by treatment. a** Each fork is assigned a raw stall score by computing the decrease in frequency of positive BrdU calls (A,B: frequencies) near the end of the fork. A nonlinear scaling is applied to create a final stall score. **b** Distribution of stall scores for untreated cells, as well as for each monotherapy. Vertical red line median, IQR interquartile range, N number of fork calls. Treatment effect is reported in Figure S7. **c** Measurements of the speed, analogue incorporation, and stall score of each fork were used to create an 8-dimensional replication stress signature, shown embedded into two dimensions using UMAP. Left: Each point is the signature for a single fork, points are coloured

according to the monotherapy delivered, and biological replicates are shown as different markers. The centroids from K-means clustering (K = 5) are shown as red boxed crosses. Right: Kernel density estimation to show distinct clusters, coloured from blue (low density of points) to red (high density of points). Clustering performance is shown in Figure S8. To account for the treatment protocol in Fig. 2a where HU was added with the BrdU pulse, Figure S9 repeats this analysis using the length of the BrdU track as the only metric of fork speed. **d** Repeat of (**c**) that excludes any measurement of fork speed in the replication stress signature. Source data are provided as a Source Data file.

genome; demonstrated that an inexpensive handheld device produces enough data to create replication stress signatures that can accurately stratify cells according to the part of the replication stress response pathway that was inhibited; and used a single sequencing run to show how replication fork speed changes across replication timing domains. The result is a major step change over the previous "gold standard" of DNA fibre analysis for measuring replication stress in cancer cells: Our method categorically supersedes DNA fibre in throughput, resolution, cost-effectiveness, and automation, all while enabling us to map stressed forks to the genome which DNA fibre analysis is fundamentally unable to do.

In addition, our method distinguishes between slow-moving and stalled forks. This distinction is critical for characterising the effect of chemotherapies on replication forks, as we have shown that HU causes the former while ATR inhibition causes the latter. We have demonstrated that the ability to map replication forks to genomic positions makes it possible to leverage pre-existing high resolution Repli-Seq

data to investigate how replication timing impacts fork movement. This can naturally be extended to datasets measuring other genomic features such as epigenetic regulation and the chromatin landscape.

The sensitivity of our approach is highlighted by its ability to identify increased fork stress and reduced fork speed in HCT116 $CDK2^{AF/AF}$ cells that proliferate with a viable disruption of the intra-S-phase checkpoint[40]. Our findings provide a quantitative measure of the increased replication stress previously described in these cells and demonstrates how restricting CDK2 activity is important for suppressing fork stress and ensuring replication fork speeds rise throughout S-phase. Our study has not accounted for any potential changes in replication timing in the HCT116 $CDK2^{AF/AF}$ and could have been improved by generating equivalent high resolution Repli-Seq timing reference for this cell line.

With future development, we anticipate further improvements to this method's throughput, utility, and scope. The 15 min analogue pulse represents 3% of an 8 h human cell S-phase, hence many of

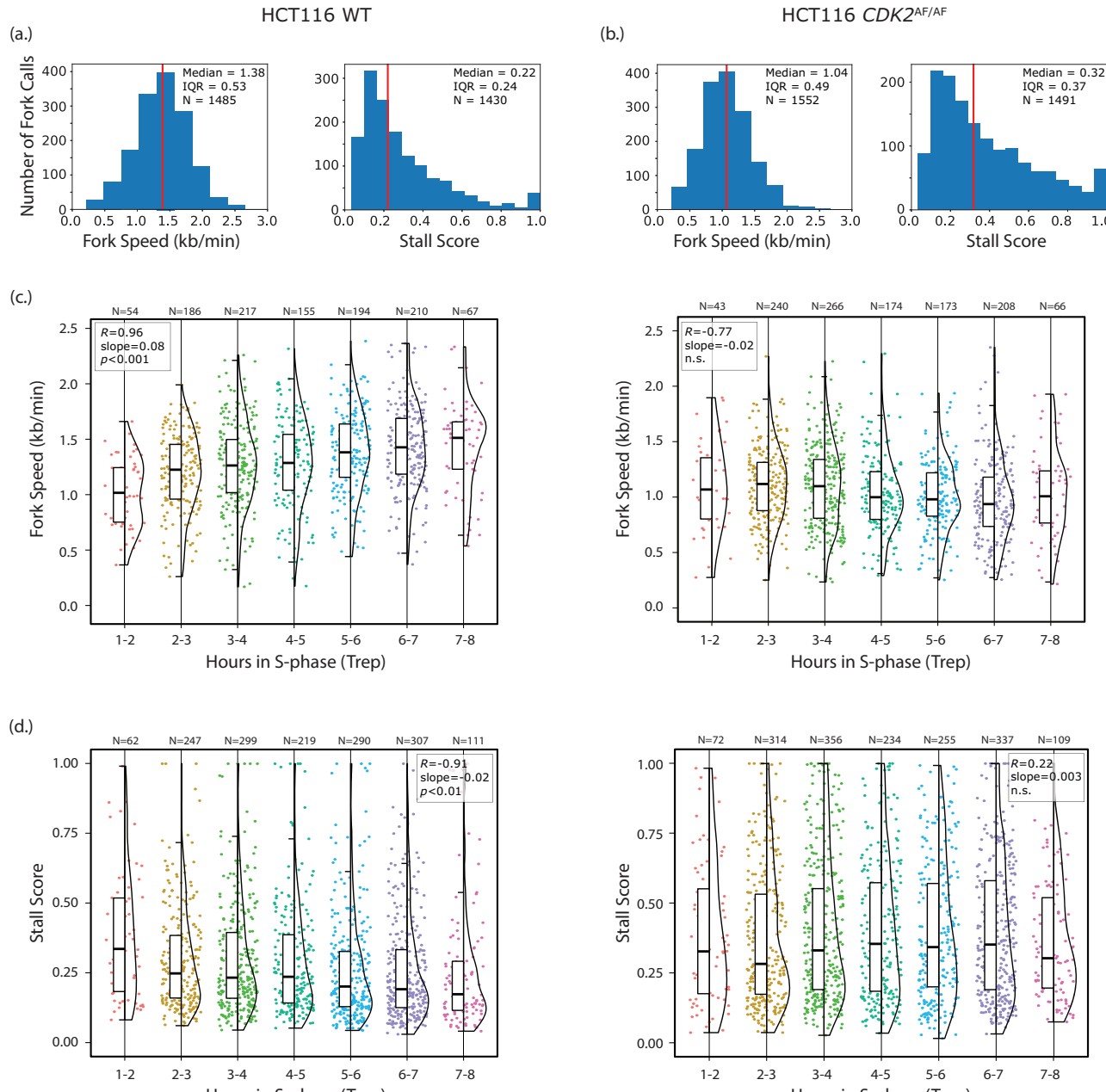

**Fig. 4 | Replication stress dynamics change over S-phase and are checkpoint-dependent.** Left column shows results for HCT116 wild-type cells and right column shows results for the HCT116 *CDK2*^AF/AF mutant. **a** Distribution of fork speeds and stall scores for HCT116 wild-type cells. **b** Distribution of fork speeds and stall scores for the HCT116 *CDK2*^AF/AF mutant. Vertical red line: median; IQR interquartile range, N number of fork calls. The equivalent analysis for a biological replicate is shown in Figure S10 and the effects of the mutation on both fork speed and stall score is reported in Figure S11. **c** Distribution of fork speeds and (**d**) stall scores of forks grouped by the median replication time (Trep) of their genomic position. Fork speeds and stall scores mapping to each hour in S-phase are shown as strip plots,

violin plots, and box plots with the horizontal black line indicating the median and boxes showing the IQR. Whisker length is 1.5·IQR. The Pearson correlation coefficient (R), slope, and p-value for each panel was computed via linear least-squares regression on the median at each time point. For fork speed, the alternative hypothesis was that the slope of the regression line was greater than zero; for stall score, the alternative hypothesis was that the slope was less than zero. The correlation was significant for both fork speed ($p = 0.00018$) and stall score ($p = 0.0020$) for wild-type cells and not significant for fork speed ($p = 0.98$) or stall score ($p = 0.69$) for the *CDK2*^AF/AF mutant. Source data are provided as a Source Data file.

molecules sequenced do not include a fork track. A pulldown of EdU prior to sequencing would enrich the percentage of reads with a fork track and would also negate the need for S-phase enrichment with reporter systems. We have also observed a slight bias towards measuring leading strand synthesis, which may be due to 3' overhangs on lagging strands reducing capture by transposase during ultra-high molecular weight library preparation. In addition, we envisage the future detection of 5-methylcytosine and 5-hydroxymethylcytosine

alongside BrdU and EdU to show replication fork movement, origin firing, and epigenetic markers on the same molecule. This study focused on three cell lines and four treatments, but the method is usable for any human cell line and is extensible to tumour organoids and suspension cell cultures. While we have used the detection of BrdU and EdU for the purposes of studying DNA replication, the accurate detection of base analogues in single molecules carries much broader utility, with an example of future work being the incorporation of base

(a.)

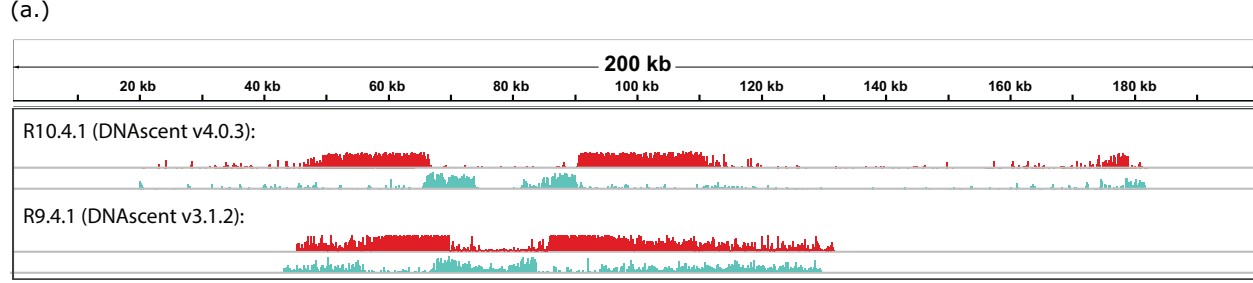

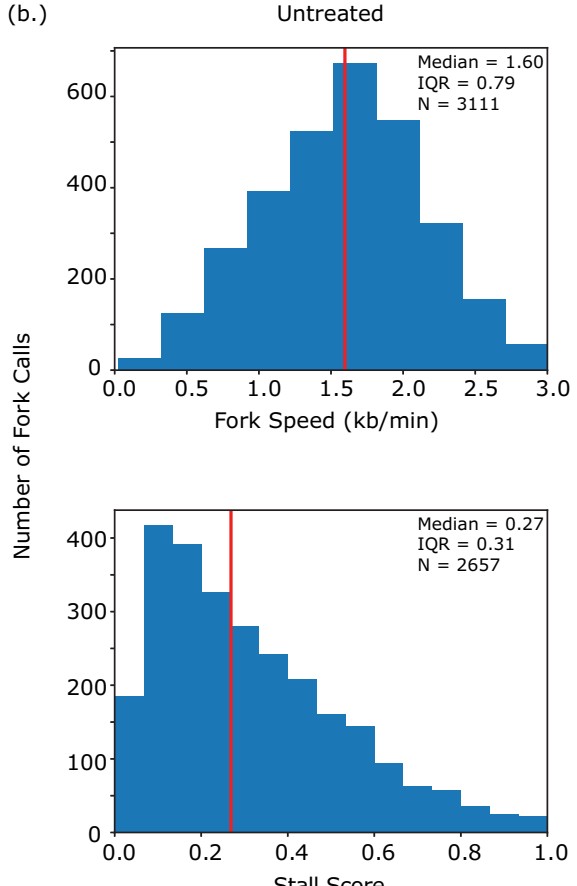

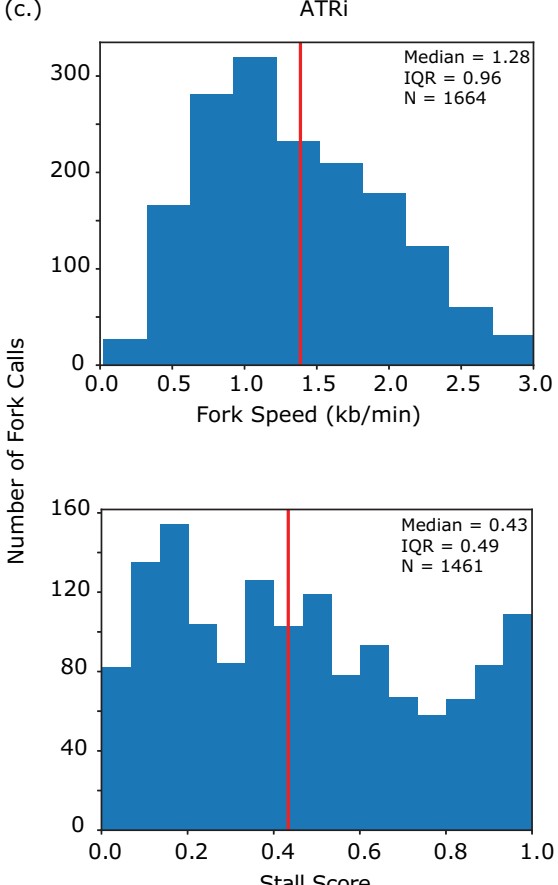

**Fig. 5 | DNAscent v4 on R10.4.1 chemistry. a** Two representative origin calls from RPE1 cells. As in Figs. 1–3, a 5-minute EdU pulse was followed by a 10-minute BrdU pulse and tracks show the probability of BrdU (red) and EdU (blue) at each thymidine position along the molecule. The upper molecule was sequenced using the R10.4.1 chemistry with BrdU and EdU probabilities annotated with DNAscent v4 while the lower molecule was sequenced using the R9.4.1 chemistry with analogue probabilities annotated with DNAscent v3.1.2. **b** Distribution of replication fork speeds and stall scores for untreated RPE1 cells and (**c**) RPE1 cells treated with the ATR inhibitor VE-821. Both samples were sequenced on the R10.4.1 chemistry. Vertical red line median, IQR interquartile range, N number of fork calls. Source data are provided as a Source Data file.

analogues during nucleotide excision repair or long-patch base excision repair.

Replication stress induced by inhibiting different parts of the RSR pathway is not created equally: the impact on replication forks will be highly dependent on the chemotherapy, dose, and timing which will become even more complex when treatments are given in combination. Our replication stress signature framework can scale with this complexity by incorporating further genomic features into the signature. This includes features from the sequenced single molecule such as mismatches and epigenetic markers, as well as population-level features such as chromatin accessibility, replication timing, and gene expression.

Our approach to developing the DNAscent software has been guided by DNA fibre assays to measure fork dynamics. We have tuned the fork detection algorithm to call the start and end of replication tracks at the peaks of EdU and BrdU detection because this is the strategy that aligns closest to DNA fibre analysis. We do not expect DNAscent to report fork speeds that are consistent with DNA fibre assays in all experimental conditions, as they are fundamentally different analogue detection strategies. For example, one of our engineering considerations was that DNAscent would find it challenging to assess fork speed and stall score in regions of weak or variable analogue incorporation. Therefore, while the increased resolution and sensitivity of analogue detection with DNAscent identifies small stretches of low-level analogue incorporation, only regions of high analogue incorporation were used to calculate fork speed (Fig. 2b). Stall scores are calculated in a manner that is unique to the nanopore sequencing platform and differences in stall scores cannot be easily

validated with another approach. While the biological explanation for the large increase in the fork stall score observed with ATRi is not completely understood, we speculate that it could be due to increased stalling or pausing of forks during ATR inhibition but cannot rule out that ATR dependent regulation of nucleotide synthesis is also a contributing factor to this phenotype[43–45].

When engineering DNAscent, our design choices were made to prioritise the accuracy of replication fork speed and stall score. This necessarily required trade-offs in other areas, particularly on the location of the boundary between the EdU and BrdU regions of a fork track. We often observe noise at the EdU-to-BrdU transition whereby EdU incorporation is elevated within the BrdU region, as well as ambiguity over where this boundary should be placed. These two phenomena are visible, respectively, in the representative ATRi and WEE1i fork tracks of Fig. 2b. Our design choices were made to minimise risk of the major segmentation error that would occur if, for example, a short segment of elevated EdU incorporation within a BrdU region was mistakenly called as an origin rather than ignored as noise. By making the calling of EdU and BrdU regions conservative to avoid these errors, DNAscent tends to leave a gap between the EdU and BrdU regions of a fork track. This can sometimes result in a ratio of BrdU-to-EdU track lengths that do not match the expected 2:1 ratio given the 5 min EdU pulse and 10 min BrdU pulse. Therefore, while we have added HU with the BrdU pulse in this study to serve as an internal control to verify shortening of the BrdU track relative to the EdU track (Figure S3), we would not generally recommend this experimental setup to test the effectiveness of an inhibitor on replication fork dynamics. We instead suggest adding the inhibitor with or before the first analogue pulse.

In summary, our method is cost-effective, automated, and provides fundamentally more layers of information to differentiate between stressed forks; all of these attributes are necessary to support high-throughput screens of how cancer subtypes respond to replication-based therapies.

## Methods

### Tissue culture, lentiviral transduction, and S-phase enrichment via FACS

A2058 cells were grown in DMEM with 10% FBS and 1% penicillin–streptomycin. HCT116 cells were grown in McCoy's 5a modified media with 10% FBS and 1% penicillin–streptomycin. pLenti-CMV-Blast-PIP-FUCCI was a gift from Jean Cook (Addgene plasmid # 138715). RRID: Addgene_138715. The PIP-FUCCI lentiviral transfer plasmid was cotransfected with psPAX2 pMD2.G into 293 T cells. 48 h later, supernatants were filtered, mixed 1:1 with fresh medium containing polybrene (8 µg/mL), and applied to target cells for 16–24 h. A2058 and HCT116 cells expressing PIP-FUCCI were labelled with 50 µM EdU for 5 min, washed 3x in PBS (taking about 1 min for each) and labelled with 50 µM BrdU for 10 mins, washed 3x PBS and chased with 100 µM thymidine for 20 min then trypsinized in 0.05% trypsin. S-phase cells were isolated by FACS sorting (mCherry⁺, mVenus⁻) using a BD FACS Aria Fusion or MoFlo Astrios EQ. Sorted cells were pelleted and stored at −80c. Drug treatments: HU (2 mM) added during BrdU pulse, PARPi Olaparib (10 µM) for 24 h, Wee1i MK1775 (1 µM) 2 h and ATRi VE-821 (10 µM) 30 min.

### General purpose S-phase enrichment

The PIP-FUCCI system was used in this study only to increase the number of fork calls per MinION flow cell; it does not otherwise affect the results. Other cell synchronisation techniques could also be used to enrich S-phase cell, such as mimosine arrest or CDK4/6 inhibition and release[46]. Cell synchronisation techniques may impact fork stall scores depending on the end user's application.

### DNA preparation

Ultra-high molecular weight (UHMW) DNA was extracted using Nanobind CBB Big DNA Kit (SKU NB-900-001-01, Circulomics) and UHMW DNA Aux Kit (NB-900-101-01, Circulomics) according to the manufacturer's protocol (Document ID: EXT-CLU-001). DNA was incubated at 37 °C for 1 h at the elution step. The DNA was eluted in 760 µL Buffer EB (6 million cells for 6x MinION library sequencing) or in 506 µL (4–5.5 million cells for 4x MinION library sequencing). To measure the recovery and the quality of viscus UHMW DNA in Qubit and NanoDrop, the DNA sample was prepared according to the method described by Koetsier and Cantor[47]. DNA purity was checked by the ratio of DNA concentration measured in NanoDrop and Qubit (1-1.5).

### Library preparation and nanopore sequencing

libraries were prepared using Ultra-Long DNA Sequencing Kit (SQK-ULK001, ONT) and Nanobind UL Library Prep Kit (NB-900-601-01, Circulomics) according to the NanoBind Library Prep - Ultra Long Sequencing protocol (Document ID: LBP-ULN-001; 03/24/2021, Circulomics). UHMW DNA library was eluted in 225 µL (6 million cells) or 150 µL (4–5.5 million cells) ONT Elution Buffer (EB; SQK-ULK001, ONT). The loading library was prepared according to the manufacturer's protocol and was loaded onto MinION Flow Cell R9.4.1(FLO-MIN106D, ONT). Sequencing run was performed using MinKNOW (v21.11.7–22.08.4) for 96 h run script with the sequencing kit set to SQK-ULK001. To load the fresh loading library, the sequencing run was paused after 24 h and the flow cell was washed with Flow Cell Wash Kit (EXP-WSH004, ONT) according to the manufacturer's guidelines.

### Basecalling and Genome Alignment

On R9.4.1, Oxford Nanopore sequencing reads were basecalled using Guppy (v5.0.11) using the dna_r9.4.1_450bps_fast configuration. Basecalled reads were aligned to the human genome using minimap2 (v2.17-r941) using the map-ont setting. Reads from A2058 cells were aligned to the chm13v2.0 human reference genome from the Telomere-to-Telomere (T2T) Consortium[48]. Reads from HCT116 were aligned to the hg38 (GRCh38.p13) assembly for consistency with the high-resolution Repli-Seq data from Zhao and colleagues[39]. On R10.4.1, Oxford Nanopore sequencing reads were basecalled with Dorado v0.7.2 using the dna_r10.4.1_e8.2_400bps_fast@v5.0.0model. Genome alignments were done at the time of basecalling using Dorado, and reads were aligned to the T2T chm13v2.0 human reference genome.

### Fork calling with DNAscent

On the R9.4.1 chemistry, Oxford Nanopore reads were analysed with DNAscent v3.1.2 which is available under GPL-3.0 at https://github.com/MBoemo/DNAscent. Each basecalled sequencing run was indexed with the DNAscent index subprogram, and the probability of BrdU and EdU at each nucleotide along each sequenced read was called using the DNAscent detect subprogram. DNAscent detect was run using a minimum mapping quality of 20 and a minimum mapping length of 20 kb. The DNAscent forkSense subprogram segmented each read into EdU- and BrdU-positive regions. To finalise the location of the start and end of the EdU track, DNAscent computes the average incorporation in the middle third of the track and trims back from the start of the track and the end of the track until it meets a part of the track with this average incorporation. The trimming then stops and this is given as the final track. DNAscent then does the same for the BrdU track. EdU and BrdU tracks are then matched into replication forks such that, for a rightward-moving fork, the replication fork track begins at the start of the EdU track and ends at the end of the BrdU track which is where BrdU incorporation starts to decrease at the start of the thymidine chase. Fork calls on each read were then matched into origins and termination sites. On the R10.4.1 chemistry, base analogue

detection was done on DNAscent v4.0.3 with the same minimum thresholds for read length and mapping quality.

## Measurement of fork speed with DNAscent

DNAscent forkSense outputs replication fork calls as bed files indicating the region of the genome each called fork moved through during the 15 min EdU-BrdU pulse. The length of each region (in kb) was divided by 15 min to compute the speed of each fork in kb per minute. These fork tracks will be shorter if (i.) the forks are from an origin that fired during the pulse, (ii.) the forks come together in a termination site during the pulse, or (iii.) the fork track runs off the end of the sequenced molecule. To present an accurate picture of fork speed, fork tracks were excluded from the analysis if they were on a read with a called origin or termination site, or if they started or ended within 3 kb of the end of the read.

## Fork stall calling with DNAscent

The number of positive BrdU calls (probability > 0.5 called by DNAscent detect) was calculated over a 2 kb window before (inside the BrdU-positive segment) the BrdU end of fork track and divided by the number of thymidine bases in the 2 kb window resulting in a fraction between 0 and 1 (denoted $B$). This fraction of positive BrdU calls divided by attempts was also calculated over a 2 kb window immediately after (outside the BrdU-positive segment) the BrdU end of the fork (denoted $A$). A raw score was computed:

$$R = \frac{B - A}{B}$$

A nonlinear scaling was applied to the raw score $R$ to compute a stall score between 0 and 1:

$$\text{stall score} = \alpha \log\left(1 + e^{\beta(R-1)}\right) - \alpha \log\left(1 + e^{-\beta}\right)$$

On DNAscent v3.1.2, $\alpha = 1.55$ and $\beta = 3$. This scaling is plotted in Fig. 3a and serves a dual role. First, it accounts for the fact that a fork which has not stalled during the analogue pulse will produce a relatively high raw score approximately between 0.2 and 0.4 and rescales this range closer to zero. Second, it creates more conservative estimates of raw scores in the 0.6-0.9 range so that only high-confidence fork pauses or stalls have stall scores near 1. Each fork call is annotated with a stall score in DNAscent v3.1.2. To avoid erroneous results, DNAscent declines to assign a stall score to forks where (i.) the fork track runs off the read, (ii.) forks that come together in a termination site during the pulse, and (iii.) forks where there is an insertion or deletion longer than 100 bp in the read inside one of the 2 kb windows. These forks were excluded from the analysis in Figs. 3–4. Note that for DNAscent v4.0.3, the cleaner BrdU and EdU calls meant that we tuned $\alpha = 2.63$ and $\beta = 1$ to produce consistent results across v3.1.2 and v4.0.3.

## Replication stress signatures with DNAscent

The following eight features were measured for each fork: (i.) the total length of the fork track (in bp), (ii.) the length of the EdU track (in bp), (iii.) the length of the BrdU track (in bp), (iv.) fraction of thymidine positions called as BrdU in the EdU segment, (v.) fraction of thymidine positions called as EdU in the EdU segment, (vi.) fraction of thymidine positions called as EdU in the BrdU segment, (vii.) fraction of thymidine positions called as BrdU in the BrdU segment, (viii.) the stall score. DNAscent v3.1.2 outputs these features as a bed file for ease of use. Each of these eight features, across all forks, was rescaled to the interquartile range and reduced to two dimensions using Uniform Manifold Approximation and Projection (UMAP)[38] with 25 nearest neighbours, a minimum embedded point distance of 0, and the Chebyshev metric. A total of 125 forks from each sequencing run (a total 250 forks per monotherapy, accounting for two biological repeats)

were used. Kernel density estimates were made using Seaborn (v0.11.2) with the default settings.

## Mapping forks to Trep

Gaussian smoothed and scaled high-resolution Repli-Seq data for HCT116 cells was accessed from NCBI GEO (accession number GSE137764) and Trep was calculated via sigmoid fitting as per the authors' instructions[39] for each 50 kb bin across the human genome. The Trep for each DNAscent fork call was computed using the Trep of the 50 kb genomic region that the fork call mapped to.

## Human training data for 10.4.1 chemistry

RPE1 cells were labelled with 50 μM BrdU for 72 hours or 50 μM EdU for 24 hours. Genomic DNA was extracted using NEB Monarch HMW DNA Extraction Kit for Cells & Blood (T3050L). Sequencing library was prepared using the Native Barcoding Kit 24 V14 SQK-NBD114.24.

## Human samples analysed with 10.4.1 chemistry

Human cells were labelled and sorted as above for R9. Genomic DNA was extracted using R9 method or NEB Monarch HMW DNA Extraction Kit for Cells & Blood (T3050L). Sequencing library was prepared using the Ultra-Long DNA Sequencing Kit V14 (SKU: SQK-ULK114) and samples were run on a MinION Flow Cell (R10.4.1) (SKU: FLO-MIN114.001).

## Parasite test data for 10.4.1 chemistry

*P. falciparum* parasites were tightly synchronised via Percoll gradient, as in our previous study[23], then incubated at 36 h post-invasion in 20 μM EdU for 1 h or 200 μM BrdU for 1 h. Parasite DNA was harvested and sequenced as above.

## Model training on R10.4.1 chemistry

Our training procedure generally follows that of DNAscent v2[20] and DNAscent v3[23]. To create pore models for EdU- and BrdU-containing 9mers on R10.4.1 chemistry, we used the DNAscent (v4.0.3) align subprogram on the BrdU-labelled, EdU-labelled, and untreated datasets to create a signal alignment between each possible 9mer and the measured normalised current as that 9mer passes through the nanopore. This resulted in a Gaussian distribution of normalised current measurements for each possible thymidine-containing 9mer, similar to the publicly available pore model for unlabelled 9mers released by Oxford Nanopore (https://github.com/nanoporetech/kmer_models/blob/master/dna_r10.4.1_e8.2_400bps/9mer_levels_v1.txt). Note that while previous work[18] used a two-component Gaussian mixture model to identify the labelled and unlabelled distributions, the analogue concentration used to prepare our training datasets was sufficiently high that we assumed the labelled distribution would be dominant. We therefore fit a Gaussian distribution to the normalised current for each 9mer to create a pore models for unlabelled DNA as well as for EdU-labelled and BrdU-labelled DNA on the R10.4.1 chemistry. We used these pore models to perform hidden Markov-based detection[18] to determine the log-likelihood ratio of BrdU and EdU at each thymidine position of each sequenced read. For BrdU-labelled and EdU-labelled reads, a call was considered positive if the log-likelihood ratio at a thymidine position was greater than 1.5. Reads were used for further training if more than 30% of thymidine positions were called as EdU or BrdU by the hidden Markov-based detection algorithm. Following the labelling procedure outlined in the supplemental information for our manuscript on DNAscent v2[20], we used the results of hidden Markov-based detection to label each thymidine position on the read (TP = 0.6; C = 0.65; TN = 0.9). The model architecture is detailed in Figure S12 and Table S3. The training data was comprised of 50,000 2-kilobase read segments for each of the EdU-labelled, BrdU-labelled, and unlabelled conditions. After training the model for 10 epochs, we used the model to call the probability of BrdU and EdU at each thymidine position across every read in the original BrdU-labelled and EdU-labelled

datasets. We set the probability threshold above which a BrdU or EdU call is considered positive at 0.5 and pulled out reads for further training if more than 30% of thymidine positions were called as EdU or BrdU. This model was used to label the reads (TP = 0.8; C = 0.8; TN = 0.95). We then followed the data augmentation procedure outlined in Figure S2 of our DNAscent v2 manuscript[20] with the number of training 2-kilobase segments for each augmentation condition specified in Table S5. This resulted in a total of 230,000 training segments, and we trained the model architecture in Figure S12 on this data for a total of 15 epochs.

## Model inputs for R10.4.1 chemistry

A key change between DNAscent v3 and DNAscent v4 is how inputs to the neural network are handled. As shown in Figure S12, the signal input is a tensor of normalised current signals that were aligned to each base on the sequenced molecule using an in-built hidden Markov alignment. At each base position, these signals are encoded by a GRU stack. To create the base sequence input, we determined that the magnitude of the signal shift was primarily governed by the middle 5mer of each 9mer. Therefore, each 9mer was decomposed into a middle 5mer and a flanking 4mer. For example, in 9mer given by bases $N_1N_2N_3N_4N_5N_6N_7N_8N_9$, the middle 5mer is $N_3N_4N_5N_6N_7$ and the flanking 4mer is $N_1N_2N_8N_9$. We then built embeddings for the middle 5mer and embeddings for the flanking 4mer. A tensor of the 5mer embeddings, a tensor of the 4mer embeddings, and a tensor of the GRU-encoded signal are concatenated and passed to the first convolutional layer.

## DNA fibre analysis

A2058 cells were labelled with IdU and CldU (50 µM) for 20 min each. DNA fibres were prepared as previously described in ref. 49. Cells were trypsinized and resuspended at $1 \times 10^6$ cells/ml in PBS. 2 µl of cell suspension was placed onto a glass slide and lysed in 10 µL of lysis buffer (200 mM Tris-HCl (pH 7.4), 0.5% SDS, 50 mM EDTA). After 6 min, the slides were tilted at 15° to allow the DNA to spread. Slides were air-dried for 30 min, fixed in methanol and acetic acid (3:1) for 2 min, and refrigerated overnight before immunolabeling. DNA was denatured with 4 M HCl for 20 min. Slides were incubated in blocking buffer (PBS + 0.1% Triton X-100 + 10% goat serum) for 1 h. IdU and CldU were detected using rat anti-BrdU (Abcam ab6326, 1:200) and mouse anti-BrdU (BD 347580, 1:200). Secondary antibody staining was performed using Alexa Fluor 488-labeled goat anti-mouse IgG antibody (Invitrogen A-11029 1:200) and Alexa Fluor 568-labeled goat anti-rat antibody (Invitrogen A11036 1:200). Slides were mounted in Prolong Gold and imaged on a DeltaVision Ultra. Replication track lengths were measured using Image J.

## Treatment effect

We employed a hierarchical Bayesian model to estimate the posterior distribution of the effect of agents/mutations on replication fork speed and stall score, as well as determine whether the effect of this treatment exceeded background variation between replicates. Fork data from each replicate was modelled using a partial pooling structure to determine within-group variability and between-group differences. For each group, replicate-level means were modelled as draws from a group-level distribution, allowing for uncertainty at both the replicate and condition levels. Replication fork speed was assumed to follow a normal distribution and weakly informative priors were placed on group means ($\mu$=1.5, $\sigma$ = 0.5 for untreated and $\mu$ = 0.7, $\sigma$ = 0.5 for treated) with half-normal ($\sigma$ = 0.5) priors on standard deviations. Stall score was assumed to follow a Beta distribution to reflect the score being bound on [0,1]. These distributions were parametrised via a group mean and shared precision. The prior on the group mean was Beta-distributed ($\alpha$ = 2, $\beta$ = 5 for both treated and untreated) and the prior on precision was Gamma-distributed ($\alpha$ = 2, $\beta$ = 0.1). Posterior

inference was done using the No-U-Turn Sampler (NUTS) which drew 2000 posterior samples after 1000 tuning steps. For guidance, we defined a Region of Practical Equivalence (ROPE) as ±0.05 for stall score and ±0.1 kb/min for fork speed. We reported a treatment effect ($\Delta_1$, the difference between group mean of treated and the group mean of untreated), control replicate variability ($\Delta_2$, the absolute value of the difference between the untreated replicate means), treated replicate variability (the absolute value of the difference between the treated replicate means), and adjusted treatment effect ($\Delta_1 - \Delta_2$). These are reported for fork speed and stall score in Figures S5, S7, and S11. We considered an effect to be significant if at least 95% of the treatment effect posterior lies outside the ROPE.

## Reporting summary

Further information on research design is available in the Nature Portfolio Reporting Summary linked to this article.

## Data availability

The Nanopore sequencing data generated in this study have been deposited in NCBI's Gene Expression Omnibus under the GEO Series accession number GSE216926 and the European Nucleotide Archive under accession number PRJEB80561. Source data are provided with this paper.

## Code availability

The DNAscent software is available at https://github.com/MBoemo/DNAscent under the GPL-3.0 open-source license[50]. DNAscent v3.1.2 was used for analysis on the R9.4.1 Oxford Nanopore chemistry while DNAscent v4.0.3 was used for analysis on the R10.4.1 chemistry. Both versions are available as Singularity images at https://cloud.sylabs.io/library/mboemo/dnascent/dnascent.

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

## Acknowledgements

The authors would like to thank Brian Gabrielli (Mater Research Institute), Valentine Murigneux (Genome Innovation Hub, University of Queensland), and all the members of the Jones and Clarke laboratories (Frazer Institute) as well as the Boemo laboratory (Cambridge Pathology) for guidance and helpful conversations. We would like to acknowledge the Translational Research Institute (TRI) for providing an excellent research environment and core facilities that enabled this research. We particularly thank Lucie Leveque-El Mouttie, David Sester, Dalia Khalil, Yitian Ding and Andy Wu from the Flow Cytometry Core Facility for invaluable help and technical assistance. The Translational Research Institute is supported by a grant from the Australian and Queensland Governments. Research by MAB is supported by the Isaac Newton Trust (19.39B), the Royal Society (RGS\R1\201251), the Leverhulme Trust (RPG-2022-028), and the Medical Sciences Research Council (MR/W031442/1). This work was supported by the Cancer Research UK Cambridge Centre (C9685/A25117) via a Cancer Research UK Cambridge Centre PhD Studentship to PLP. MJKJ and PRC are supported by the Australian Research Council grant DP210102704 and the Genome Innovation Hub at the University of Queensland. MJKJ is also supported by funding from the University of

Queensland FACULTY OF SCIENCE BIRRST PARTNER Scheme. F.I.G.T. was supported by the European Research Council under the European Union's Horizon 2020 research and innovation programme (ERC-2016-COG25126 to C.J.M.)

## Author contributions

M.J.K.J., S.E.M. and M.A.B. conceptualised and designed the study. M.A.B. and P.L.P. developed the software and did the bioinformatics analyses. S.K.R., M.J.K.J., and A.B.M. conducted the wet lab research and generated nanopore sequencing for human cell lines. FIGT generated the nanopore sequencing data for *P. falciparum*. J.K.P. provided reagents and intellectual contributions. M.J.K.J., M.A.B., S.E.M., C.J.M., and P.R.C. acquired funding for the project. M.A.B., M.J.K.J. and A.B.M. made the figures. M.J.K.J. and M.A.B. supervised the project and wrote the manuscript with input from all authors.

## Competing interests

P.L.P. has received small grants from Oxford Nanopore Technologies to cover travel and fees associated with presenting research at scientific meetings. The remaining authors declare no competing interests.
