## [Transparent Peer Review file · Nature Communications]

A high-resolution, nanopore-based artificial intelligence assay for DNA replication stress in human cancer cells

Corresponding Author: Dr Michael Boemo

Version 0:

Reviewer comments:

Reviewer #1

(Remarks to the Author)

In this manuscript, Jones and colleagues use an advanced nanopore sequencing-based replication mapping technique to extensively characterize DNA replication stress induced by various chemotherapeutic agents in human cancer cells. Thanks to a novel version of their homemade DNAscent software now allowing the detection of two thymidine analogues, namely ethynyldeoxyuridine (EdU) and bromodeoxyuridine (BrdU), sequentially incorporated into replicating DNA, the authors measured the speed of thousands of replication forks in human cancer cells and revealed chemotherapy-associated "replication stress signatures".

In recent years, nanopore sequencing and artificial intelligence have started to revolutionize DNA replication analysis in eucaryotes, and the study by Jones and colleagues is a new illustration of this. To the best of my knowledge, this is the first time this approach is applied to the human genome. Moreover, Michael Boemo's team is the only one mastering nanopore analysis of dual EdU/BrdU pulses, which holds great potential to inspect fork slowing and stalling associated with replication stress, as underlined here. In this regard, the uncovering of putative "replication stress signatures" for different chemotherapies is new and promising. Unfortunately, several major claims of this study are insufficiently supported by the data and although the manuscript is concise and easy to read, this comes at the expense of a thorough assessment of the methods described here.

Major concerns

As anticipated, the authors' validation process relies on the ability of their method to retrieve previously reported fork speed estimates and changes in fork velocity upon specific treatments. Although they do recover several existing results (fork slow down upon ATR or WEE1 inhibition, fork acceleration in the presence of a PARP inhibitor), certain comparisons are too imprecise to conclude. In addition, most data about fork stalling evidently differ from what is known in the literature, which the authors do not comment but which inevitably questions their approach.

Among the data contradicting published results is the measurement of fork speed in the presence of HU. I was positively astonished by the fact that the concentration of HU used in this study (2 mM) resulted in such a low slowing down of the forks according to DNAscent (0.75 kb/min versus 1.5 kb/min in untreated cells). Indeed, it has been reported in U2OS cells that fork speed is about 100 bp/min after a 30 min treatment with 1 mM HU, versus 1.25 kb/min in untreated cells (Somyajit et al 2017), and that fork velocity is so low after a 1 hour treatment with 1 mM HU that it "cannot be measured" in human lymphoblasts (Koundrioukoff et al 2013). It has also been reported that fork speed is drastically reduced 15 min after addition of 3 mM HU (Dungrawala et al 2015), ie in conditions that resemble those used in this manuscript (however, it is important to note that in Dungrawala's study, forks progress at 540 bp/min in untreated cells, which is 3 times slower than in A2058 cells). Although results from Jones and colleagues might be explained by a low sensitivity of A2058 cells to HU (are there reasons to believe so?) or by experimental differences with Somyajit, Koundrioukoff and Dungrawala's studies, they are so unexpected that I can only but doubt them for now, and presume that they may be due to an inaccurate estimation of fork speed by DNAscent. A direct comparison with fork speed measured by standard DNA fiber analysis in the exact same conditions of HU treatment would greatly help clarify this point. DNA fiber analysis has biases of its own and will not give the "absolute truth" regarding the effect of HU on DNA replication in A2058 cells, but it will assuredly allow to determine if 2 mM HU has a major or more subtle impact on fork velocity. One last comment: although I cannot see any rationale for this, could it be that the authors actually pooled the measurement of fork progression during the EdU pulse in the absence of HU with that during the BrdU pulse in the presence of HU to calculate HU fork speed, resulting in such a high velocity? The read

presented in Figure 2b does not suggest it since the length of the BrdU-positive and EdU-positive regions is approximately the same, which is compatible with a 50% reduction in fork speed after HU addition. However, this read may not be representative, and there is no mention in the “Online Methods” that HU data were treated differently from the other samples, so it remains possible that HU fork speed was indeed determined by dividing the length of the EdU/BrdU region by 15 minutes. The authors must make this point crystal clear.

I was also very surprised that, according to DNAscent stalling score, HU treatment “resulted in a similar distribution of stall scores to that of untreated cells”, whereas HU is basically the prototype of fork stalling inducers (see for instance Techer et al 2013, Cali et al 2016, Saxena et al 2018, Datta et al 2021). Similarly, it has been reported twice using DNA fiber analysis that cells deficient for ATR exhibit a normal rate of fork stalling (Speroni et al 2012, Koundrioukoff et al 2013), in contrast with the results depicted here (“treatment with ATR inhibitors resulted in a marked increase in stall score indicative of frequent fork stalling”). It is possible that ATR inhibition leads to different outcomes in terms of DNA replication than ATR silencing used by Speroni and Koundrioukoff, as reported for PARP (Maya-Mendoza et al 2018) and, once again, I am not stating that DNA fiber assays provide unsurpassable results in terms of fork stalling analysis. Still, DNAscent stalling results for HU and ATR are so different from previous studies that they make me question their soundness, especially since there is no other validation of the stall score in the manuscript than the comparison with published results. The seeming reproducibility of stall scores is not that reassuring since the authors merely claim that results “were consistent across biological replicates” instead of performing a bona fide statistical analysis, which is mandatory to conclude (statistical tests must also be performed to compare fork speed between the different samples).

A frequently used marker of fork stalling in DNA fiber analysis utilizing dual pulse labelling is the ratio between the length of the tracks synthesized during the first and second pulse, commonly referred to as “fork asymmetry”. In this regard, I am intrigued by the reads presented in Fig. 2b. The ratio between the length of the EdU- and BrdU-positive regions seems to be approximately 2 for untreated cells, as expected; quite surprisingly however, this ratio is also near 2 for the ATRi read, but far from 2 for both the WEE1i and PARPi reads, which is in complete contrast with the stall scores. Of course, the single read presented for each condition cannot be considered as representative, but I believe a global analysis of the EdU/BrdU ratios would be informative and important to challenge the stall scores. The authors could additionally compute the ratio between the length of sister forks present on the same read, that is also used to assess fork stalling, and see if the results are in line with those of the stall scores or not. Perhaps using synthetic signals as in Theulot et al 2022 would be an interesting approach to test both fork speeds and stall scores output by DNAscent. In conclusion, the stall score must be fully validated before being used as a trustable marker of fork stalling. Because of this, I am far from convinced that the authors' method can differentiate between fork slowing and fork stalling, as alleged in the manuscript based on ATR and HU stall scores. The uncertainties about the meaning of the stall score also cast a shadow on the different “replication stress signatures” detected for HU, ATRi, WEE1i and PARPi, as they seem to be largely based on this parameter. Speaking of the UMAP analyses, I also wonder why the untreated sample was not included.

Finally, methods are insufficiently detailed in the manuscript. Notably, BrdU and EdU calling in human reads using DNAscent is barely described. DNAscent software development and model training for *P. falciparum* DNA is presented in a recent bioRxiv paper from Boemo's team (Replication origin mapping in the malaria parasite *Plasmodium falciparum*), but what about human DNA? Since no human datasets for EdU and BrdU model training are described, I assume that the model trained with *P. falciparum* DNA was used in this study. If so, please provide the rationale for this given the differences between the human and *P. falciparum* genomes, particularly in terms of GC content. Otherwise, please explain the training procedure. In addition, please show how the reliability of EdU vs BrdU vs thymidine detection in human reads was validated. The “Fork calling with DNAscent” paragraph should also extensively describe how the start and end of replication forks were determined, specifying how EdU- and BrdU-positive regions within a read were segmented by forkSense. The EdU background signal seems quite high on reads presented in Figs. 1f and 2b, so how was the “start” positioned? Similarly, when the authors state that “replication fork track ends where BrdU incorporation starts to decrease at the start of the thymidine chase”, how was this located? It is apparent in Fig. 2b that local decreases due to signal noise are present within BrdU-positive regions, but have been ignored, rightfully or not, by the segmentation algorithm. In fact, EdU and BrdU-positive regions are not necessarily in agreement with the visual aspect of the EdU and BrdU profiles for reads shown in Fig. 2b, and they are often far from being contiguous although the authors seem to consider that the washing time between EdU and BrdU pulses is negligible (see below). I would like the authors to comment on that, too. In conclusion, the authors must report how the whole fork calling procedure was validated. Such details are critically important to evaluate forkSense accuracy. The need for an in-depth description also applies to the “Fork stall calling with DNAscent” paragraph. As stated above, I truly wonder how the fork stalling scoring procedure was validated apart from comparing with previous biological results. How was the non-linear scaling “approved”, for instance?

Other points to be addressed:

- To the best of my knowledge, there is no previous estimate of fork speed in A2058 cells. DNAscent mean fork speed of 1.5 kb/min is clearly within the range of fork speeds typically found in human cells, but there is no guarantee that this value is accurate since fork velocity can show great variability between cell lines, or even within a given cell line. A comparison with fork speed measured by standard DNA fiber analysis in this cell line cultured in the exact same conditions would help assess the accuracy of DNAscent estimation. Similarly, fork speed estimated in HCT116 cells by DNAscent is 20% less than what has been found by DNA fiber analysis (Fu et al, 2015), which is no insignificant difference to me. This is most likely attributable to slight differences in technical/experimental conditions between laboratories, but it could also reveal a certain inaccuracy of DNAscent. Like for A2058 cells, a measurement of fork velocity by DNA fiber assay in HCT116 cells by the authors would help provide an answer. The whole manuscript would in fact greatly benefit from a systematic comparison between fork speed/stalling estimated either by DNAscent or by standard DNA fiber analysis for all cell lines and culture conditions.

- Please specify the duration of the wash between the EdU and BrdU pulses, which is not taken into account in the total time used to calculate fork speed. Three washes with PBS may take about 30 seconds to me, which I guess could be considered negligible compared to 15 minutes. Still, I wonder if slightly different wash durations from sample to sample may impact signal segmentation and contribute to variations in the length of the space separating EdU- and BrdU-positive regions, as seen in Fig. 2b.
- In Fig. 4c and 4d, creating discrete Trep categories may induce visual artifacts. A statistical analysis is required to sustain the authors' conclusions.
- The replication timing may be different in HCT116 CDK2AF/AF mutant compared to WT cells, especially since this mutant displays aberrant replication dynamics (Hughes et al, 2013). The authors either need to provide evidence of similar timing in both cell lines or tune down their conclusions.
- How is fork speed estimated by DNAscent affected by read length?
- The use of the mean as a parameter to describe fork stall scores, which do not have a normal distribution, is not the best choice.
- The authors use "DNA combing" to indiscriminately refer to molecular DNA combing or DNA fiber spreading. Please use "DNA fiber analysis" instead.
- "Long-read sequencing using the Oxford Nanopore Technologies (ONT) platform has enabled the detection of replication origins and fork movement in budding yeast¹⁸⁻²², the malaria parasite *Plasmodium falciparum*²³, and human mitochondrial DNA²⁴": the claim that it is possible to detect replication origins and fork movement in human mitochondrial DNA seems an over-statement of Georgieva and colleagues' conclusions in reference 24; please rephrase.
- "As shown in *S. cerevisiae*^{18, 20, 21}, fork stalling manifests as a sharp drop-off in analogue incorporation into nascent DNA": please remove reference 21, which has not explicitly shown this.
- Figure 2 caption: replace (d.) by (c.).
- "Forks measured from cells treated with each chemotherapy clustered together, showing that disrupting different elements of the DNA replication stress response (RSR) pathway creates a replication stress signature in this 8-dimensional space (Figure 3c): please remove "RSR", which has already been defined.
- "We have demonstrated the ability to map replication forks to genomic positions makes it possible to leverage pre-existing high resolution Repli-Seq data to investigate how replication timing impacts fork movement": please add "that" after "demonstrated" to read "We have demonstrated that the ability".
- "We have also observed a slight bias towards measuring leading strand synthesis, which may be due to 3' overhangs on lagging strands reducing capture by transposase during ultra-high molecular weight library preparation": these data should be presented in the manuscript as it might affect some results.
- "The method is usable for any cancer cell line and is extensible to tumour organoids": the labelling protocol described here is readily applicable to adherent cells, but what about cells growing in suspension culture? Please comment.
- Online methods, DNA preparation paragraph: please define "UHMW".

Reviewer #2

(Remarks to the Author)

In this succinct and accessible manuscript, Jones and colleagues demonstrate that nanopore sequencing of long DNA molecules sequentially labeled with two thymidine analogs allows quantitative analysis of replication-fork speed and stalling through the human genome. The authors benchmark their technique using a variety of replication stressors, show that different stresses lead to distinct, resolvable signatures of replication-fork movement, and finally provide data that indicate variable replication speed and fork-stalling propensity in early-replicating versus late-replicating regions of the genome.

Overall I am very positive about this work. As stressed by the authors, this technique (even at its current relatively limited sequencing depth) is poised to supersede DNA fiber analysis as the primary means to assay replication-fork speed and stalling *in vivo*. This manuscript will therefore be of considerable interest to the community.

I have a couple of minor comments, as well as some suggestions for additional analyses that may be possible with the authors' existing data: of course, it is possible that these analyses may be the subject of more in-depth follow-ups. I don't think these extensions are essential, but I do think that they would further strengthen the manuscript.

1. The authors claim that their technique can approach base-pair resolution. Although the absolute number of fork stalling events detected is fairly low, is it high enough to make any comparisons beyond replication timing? For example, are stalls detected primarily within genes (and if so, is the position of the stall biased in any way)?

2. Fork asymmetry is one of the more commonly used metrics in fiber analysis, and recent work from the Whitehouse lab (PMID: 35240057) indicates that sister replisomes are not coupled in budding yeast. Is the sequencing depth here sufficient to conduct an analysis of fork asymmetry and coupled/uncoupled stalling?

3. Related to point 2 above, is it possible to conduct a simple analysis of the number of replication forks detected per unit input DNA, as a proxy for origin density?

4. Using the UMAP approach outlined in figure 3, is it possible to identify a perturbation (or provide statistical probabilities) from a blinded sample?

Minor comments.

1. Fig. 1f – as noted in the figure legend, the tracks represent reads moved to the same axis from different regions of the genome: it would be helpful to make this clearer in the figure itself, perhaps by annotating the three regions.

2. I think it would be helpful to have a little more description of how stall score is calculated, and how (if at all) it is affected by variability and variation in replication-fork speed within the main text.

3. ATRi looks more variable on the UMAP than other treatments, at least to my eye. Is there a rationale for the fact that the clustering is not particularly tight? Is this just related to the variability in stalling? It would be helpful to clarify.

4. Figure 4; some basic statistical analysis of the differences in fork speed and stalling in the different replication timing bins would be helpful.

5. Some of the samples have very low fork numbers (e.g. ~50 forks for one Olaparib replicate), but the data seem quite robust. Have the authors ascertained the minimum read depth required for specific analyses?

Reviewer #3

(Remarks to the Author)

Jones et al report the use of nanopore sequencing to measure fork speed and stalling in human cancer cells. Their results outperform other previous assays and are thus an important contribution to the field. There are currently many efforts to use long-read and other sequencing approaches to assay replication dynamics, and this paper provides an elegant, powerful and well-presented approach in this respect. I have several comments/suggestions, most of them minor.

- The authors use the PIP-FUCCI system to isolate S phase cells, however can they comment on whether this is necessary for obtaining the information reported by their method or is this rather a means of reducing costs by only sequencing the relevant cells (and if so, is this the only way to enrich for S phase cells or could alternative ways be used in different cellular systems)?

- End of intro: "up to single-nucleotide resolution"- where is that shown? I'm not convinced this is the resolution the method provides with sufficient confidence.

- Can the authors explain the rationale for the different treatment durations with different drugs (figure 2a)? Is it possible that the different treatment regimens explain the clustering in figure 3c-d?

- Can the authors show one or more representative genomic regions with the actual chromosomal coordinates? Ideally the same region for different treatments. Currently "anonymous" regions are shown, raising the question of how these patterns differ across regions that vary in chromatin structure, gene density, GC content and other parameters. The actual locations of the regions shown should also be indicated in the figure legends. In figure 4 they stratify the data by replication time, however they haven't addressed whether these patterns also depend on the other factors mentioned here (or others). (On that note, showing tracks in different times along S phase is one example of how showing specific genomic regions could benefit the presentation here)

- The figures are of very high quality, however the explanations in the main text are rather brief and general. The paper will benefit from a more detailed description of the experiments and results, and there doesn't seem to be a length problem in the current version. As one (of many) notable examples, the title of the paper mentions "artificial intelligence assay" but nothing is noted in the main text about AI, making the computational methods in general (and with them, the novelty and power of the assay) rather hidden in this presentation.

- Different chemotherapies create distinct replication stress signatures: can the authors elaborate on the specific stress form(s) that each drug induces and that lead to the clustering patterns reported in figure 3?

- Figure 4c-d: the trend is somewhat difficult to see (especially in panel d). A line connecting the bar plots for each time point could make this figure more clear, although this is an option that I would leave to the discretion of the authors.

- Discussion (minor comments): "the method is usable for any cancer cell line and is extensible to tumour organoids"- is it not also applicable to non-cancer samples, such as stem cells or other cell lines? Or are these more difficult to implement in such an approach because of the need to introduce fluorescence markers and nucleotide analogs? A comment along these lines could be useful. "a natural example being a new assay for DNA repair"- can the authors elaborate on this? This isn't clear. "Our software"- as mentioned above, the attributes of the software aren't described in detail in the main text, making it difficult to fully appreciate the novelty of this study. More generally, the discussion seems to focus on the strengths of the method, but seems thin on the specific findings reported in the paper (e.g., acceleration of fork speeds along S phase, different effects of different chemotherapies, and more)- these are interesting findings and the paper would benefit from some

further discussion of them.

Version 1:

Reviewer comments:

Reviewer #1

(Remarks to the Author)

Instead of providing a point-by-point response to my comments, the authors basically cherry picked those they chose to answer to, presenting them in a sometimes incorrectly summarised way, while they simply disregarded other remarks. My concerns regarding the meaning and reliability of the fork "Stall Score", which is central to the so-called "replication stress signature" of different chemotherapies, were not properly, if at all, taken into account. Yet validating these metrics is all the more important since, whereas nanopore analysis of dual EdU/BrdU pulses presented in this manuscript was clearly new in late 2022, it has now been published twice by the authors (although uniquely in Plasmodium species and not in human cells), with the "Stall Score" and "replication stress signatures" being the real novelties here. Notably, I can't seem to find the histograms of the EdU/BrdU ratios for each sequencing run that the authors claim to show in Figure S10 (which only present sister fork ratios), not to mention that the fork asymmetry calculation that they used is incorrect. Data were certainly inverted for RPE1 and A2058 cell lines, both in Figure 1 and in the text, and I still wonder what is the actual concentration of hydroxyurea (HU) used in this study (2 mM HU, as written in the Methods section of both the original and revised manuscripts, or 200 μ M HU, as indicated by the authors in their rebuttal letter?). Finally, my questions regarding the accuracy of fork speed measurements by DNAscent still remain. Nonetheless, an important feature of this revised version is the adaptation to Oxford Nanopore Technologies' R10.4.1 chemistry.

Major concerns:

1) Whereas I previously expressed strong concerns regarding the Stall Score, stating that « « the Stall Score must be fully validated before being used as a trustable marker of fork stalling », there is no additional validation in the revised version of the manuscript, since the EdU/BrdU ratios I asked for are nowhere to be found in Figure S10 despite the authors' claim. Actually, the sister fork analysis did not show any significant difference between the treated and untreated samples, in contrast to the authors' conclusions for ATR inhibition using the Stall Score, but the number of values (which the authors must provide in Figure S10) was apparently low and possibly insufficient. No such limitations should apply to the EdU/BrdU ratios, that I once again demand to be computed.

In Figure S10, the quantity $|(left\ fork\ track\ length)/(right\ fork\ track\ length) - 1|$ is not accurate to compute fork asymmetry. For instance, if a left fork track length is twice that of the right one, the asymmetry score will be $abs(2-1)=1$, but if the right fork track length is twice that of the left one, the asymmetry score will be $abs(0.5-1)=0.5$, whereas it should be 1 in both cases. The ratio between the longest and shortest fork track lengths must be used instead.

I wonder why the number of fork calls for the Stall Score is different from the first version of the manuscript in Figures 3b, 4a, S5 and S6a, b.

As previously written, the authors must sustain their claims regarding similar or dissimilar Stall Score distributions with an appropriate statistical analysis.

2) In my earlier comments, I did not only « queried (the authors') treatment protocol with hydroxyurea, particularly when the treatment was given », but pointed out that the weak fork slowdown measured after a 2 mM HU treatment was in sharp contrast with the expected drastic decrease in fork speed in the presence of such a high drug concentration, as recurrently reported in the literature. Now that the authors have explained, in Figure S3, how fork speed with HU was computed, I must stress again that they cannot take into account both the EdU (« rest of the fork ») and BrdU tracks to calculate fork velocity with HU, since they are mixing two different experimental conditions. Only the BrdU track, ie, the portion of the signal obtained in the presence of HU, must be used. The way the « replication stress signature » was calculated for HU must also be modified accordingly.

Given the BrdU track length difference observed between untreated and HU-treated conditions in Figure S3, speed is apparently reduced by a factor of 3 with HU. This is somewhat compatible with a treatment with 200 μ M HU, the concentration indicated by the authors in the rebuttal letter. However, forks would still be moving anomalously fast if cells had been exposed to 2 mM HU, the dose indicated in the Methods section of both the original and revised manuscripts. So I wonder: what is the actual concentration of HU used in this study?

3) Commendably, in this revised version, the authors assessed fork speed in an additional cell line, namely RPE1, finding a fork speed consistent with previous studies. However, I am puzzled by the difference between DNA fiber spreading and DNAscent results for A2058 cells, which gave fork speed estimates of 1-1.13 kb/min and 1.46-1.52 kb/min, respectively. In other words, forks appear to move 30-50% faster in A2058 cells when their speed is measured by DNAscent instead of DNA fiber spreading, which is far from insignificant. Although none of these two approaches can be considered as giving the « ground truth », the authors uniquely blame DNA fiber analysis for these discrepancies without questioning their own method, which I think they should.

Previously, since I was not necessarily convinced by DNAscent's segmentation of the reads into EdU- and BrdU-positive regions, I asked the authors to « report how the whole fork calling procedure was validated » since « such details are critically important to evaluate forkSense accuracy ». According to the data presented in Figure S3, I feel that my doubts were well-founded. Indeed, it appears that, in untreated cells, BrdU tracks are 3 times longer than EdU tracks (« rest of fork »), whereas a ratio of 2 is expected given that BrdU labelling time is only twice that of EdU. Such an overestimation of BrdU track length alone could almost entirely explain the difference in speed with the DNA fiber analysis. A possibly biased evaluation of BrdU track length does not really come as a surprise in view of the exemplary reads shown in Figs. 1 and 2, since positioning the end of the fork track « where BrdU incorporation starts to decrease at the start of the thymidine chase» do seem more complicated than positioning the start of the EdU track or the EdU/BrdU transition. Because the Stall Score is also based on the analysis of the BrdU level within BrdU tracks, this issue must be thoroughly addressed.

4) Regarding the duration of the wash, I previously anticipated it to be negligible compared to the 15 minutes of EdU/BrdU labelling time. In fact, washes last for 3-4 minutes according to the authors, and I therefore wonder why fork speed is currently computed by dividing the length of the EdU/BrdU labelled region by 15 minutes and not 18-19 minutes, which is the actual time separating the start of the EdU pulse and the start of the thymidine chase. This likely results in an overestimation of fork speed although, obviously, modifying the time interval will only impact « absolute » speed values and does not prevent comparisons between untreated and treated conditions.

5) The authors are requested to respond to any comments from my first review that they have not yet addressed.

Other points to be addressed:

- Update reference 23.

- « as well as the malaria parasites *Plasmodium falciparum*²³ and *Plasmodium knowlesi*²⁴ »: please add a reference to the article « The genetic landscape of origins of replication in *P. falciparum* » by Casilda Muñoz Castellano et al, *Nucleic Acids Research*, 2024.

- « replicates with fork speeds of ~1.5 kb/min in RPE1 cells »: please replace 1.5 with 1.4 since RPE1 and A2058 cell lines have certainly been inverted in Fig. 1e.

- « a difference of approximately 0.2-0.3 kb/min »: please replace 0.2-0.3 with 0.3-0.5 since RPE1 and A2058 cell lines have certainly been inverted in Fig. 1e.

- « This study focused on two cell lines »: please replace two with three.

- Figure 1e: A205 and RPE1 cell lines have certainly been inverted; please correct.

- Figure 3c: please show the centroids for each biological replicate separately. The untreated condition is hardly clustering; could the authors explain why? (see also my related comment regarding Figure S7).

- Figure 5b caption: Distribution of replication fork speeds AND stall scores.

- Figure S3a caption: remove « was measured ».

- Figure S6 caption: Biological replicate OF the experiments shown in Figure 4.

- Figure S8 caption: Supplemental Table S3 shows the values OF the number of filters.

- Figure S7: the clustering of the untreated sample is not very convincing, since it is called more like an HU-treated sample. This is possibly related to the issue mentioned above concerning the way the HU sample was handled, with part of its signal corresponding to untreated conditions. Also, to really show « how good the clustering is », as the authors state, it seems important to perform this analysis independently for both replicates of each category.

(Remarks on code availability)

Reviewer #2

(Remarks to the Author)

The authors have done a nice job responding to critiques raised in the first round of review. This timely manuscript will be of interest to the field.

(Remarks on code availability)

Reviewer #4

(Remarks to the Author)

The authors have addressed my concerns, although I am still not fully convinced that “up to single-nucleotide resolution” is the most appropriate terminology given the authors' response; providing context to that specific choice of words would be helpful

(Remarks on code availability)

Version 2:

Reviewer comments:

Reviewer #1

(Remarks to the Author)

I thank the authors for their long and honest answer regarding the limitations of this study. I still have my doubts regarding the stall score, but research necessarily involves some uncertainty. However, only one discussion paragraph has been added to the main text regarding these limitations, and I think that readers would benefit from some of the information provided by the authors in their rebuttal letter being directly incorporated into the manuscript (I am aware that peer review files are public, but they are inevitably less consulted than the article itself). Several important additional analyses have also been performed in the revised version, which is commendable.

Minor comments:

1. “This relationship vanished in the CDK2AF/AF mutant, as these cells maintained a constant slow fork speed and high stall score across regions of the genome that replicated both early and late in wild-type cells”: even if the wording is now correct, I still wonder what this really means given that replication timing may be different in CDK2AF/AF than in WT cells.
2. The sentence “While we observed a higher variability... that exceeded this variability (Figure S11)” lacks clarity and may be rephrased.
3. “Stress score” is sometimes used in this manuscript instead of “stall score”; this may be homogenised.

(Remarks on code availability)

Reviewer #1

The reviewer raised the point of how accurate and reliable the DNAscent results are, with a particular focus on the fork speed in A2058 melanoma cells and how our results compare to that of DNA fibre.

We agree that this is a valid concern given that A2058 cells have not yet been characterised by DNA fibre. To address this, we performed three DNA fibre experiments. First, we performed DNA fibre on A2058 cells and compared the fork speeds from DNA fibre to the fork speeds from DNAscent in the new Supplemental Figure S2. While both replicates of DNA fibre were slower than what we obtained with DNAscent (1.1 kb/min and 1.0 kb/min from DNA fibre; 1.5 kb/min from each of the two DNAscent replicates), as the reviewer says, fibre cannot be taken as a gold standard of fork speed. Second, we did two additional sequencing runs in human RPE1 cells (untreated with any agents) and analysed the results with DNAscent to provide a good standard of comparison in a well-characterised cell line. These were added to Figure 1e and, as shown in that figure, DNAscent-measured fork speeds were 1.3 kb/min and 1.4 kb/min. This is in good agreement with fork speeds measured by DNA fibre (1.4 kb/min in Chen, et al. *Molecular Cell* 2015; 1.5 kb/min in Benedict, et al. *Developmental Cell* 2020; 1.5 kb/min in Pennycook, et al. *Nature Communications* 2020). Moreover, the DNAscent-measured stress scores in RPE1 cells are low which agrees with expectation.

The reviewer queried our treatment protocol with hydroxyurea, particularly when the treatment was given.

As shown by Figure 2a, we treated cells first with EdU for 5 minutes and then treated cells with 200 uM of HU together with BrdU. However, we agree that we should have made the logic for this more explicit. The purpose of this protocol was to stress test the accuracy of the segmentation such that adding HU together with BrdU should result in a shortening of the BrdU track but not the EdU track. We created Supplemental Figure S3 showing that this is the case. We added the following text to the manuscript: *“To test the method further and provide an internal control, we added HU with the BrdU pulse and verified shortening of the BrdU track but not the EdU track (Supplemental Figure S3).”*

“A global analysis of the EdU/BrdU ratios would be informative and important to challenge the stall scores. The authors could additionally compute the ratio between the length of sister forks present on the same read, that is also used to assess fork stalling, and see if the results are in line with those of the stall scores or not.”

We agree that this analysis would be informative, and we have created histograms of the EdU/BrdU ratios for each sequencing run in Supplemental Figure S10. The ratio between the lengths of sister forks present on the same read was addressed in our response to Reviewer #2 who had a similar query.

The reviewer raised questions about the model training procedure, as well as the reliability of BrdU and EdU detection in human cells.

For the R9.4.1 chemistry, the model is the same one used in our previous publication (<https://doi.org/10.1093/nar/gkad093>) which we have cited. This work includes benchmarks on BrdU and EdU detection performance and Supplemental Table S2 gives an indication of false positives in unlabelled DNA. No new model training was included in this work for the R9.4.1 chemistry, although the software was updated to detect fork stalling which we detailed in the methods section. We have added a new section detailing model training for the new R10.4.1 chemistry which is an important contribution of this manuscript. The R10.4.1 model's architecture is detailed in Supplemental Figure S8 and benchmarks are in Supplemental Figure S9 and Supplemental Table S4.

The reviewer requested that we add the untreated sample in the UMAP analysis.

We have done so in Figure 3 and we have also added metrics in Supplemental Figure S7 showing how good the clustering is (see the response below to Reviewer #2's suggestion).

The reviewer raised questions about the duration of the wash, and whether this was likely to impact the segmentation of BrdU and EdU calls into track lengths.

The PBS washes require ~1 mins each (4x T175 flasks) so between EdU and BrdU plus the total time for 3x PBS washes is 3 -4 mins. We do see examples of forks with gap and no gaps between labels. This is also true for DNA fibre methods. The time could be a factor, but it doesn't explain why gaps between labels are only seen with some forks.

The reviewer raised a concern about discrete Trep categories and whether these changes were statistically significant.

We have added the results of a one-sided Mann Whitney U test to Figure 4c and Figure 4d showing that these changes are statistically significant. We have also added corresponding text to the legend of Figure 4: "Statistics shown are from a one-sided Mann-Whitney U test (n.s.=not significant; *= $p < 0.05$; **= $p < 0.01$; ***= $p < 0.001$)."

The reviewer mentioned that mean was not the best summary statistic to use throughout the paper.

We agree, have taken this on board, and moved to median and inter-quartile range for all relevant figures.

“The authors use “DNA combing” to indiscriminately refer to molecular DNA combing or DNA fiber spreading. Please use ‘DNA fiber analysis’ instead.”

This has been fixed throughout the manuscript.

The reviewer asked us to clarify the finding of two references: Georgieva et al. and Theulot et al.

We appreciate the reviewer pointing this out and have fixed the discrepancy.

The reviewer asked us to define “UHMW”.

This has been fixed: *“Ultra-high molecular weight (UHMW) DNA was extracted using Nanobind CBB Big DNA Kit (SKU NB-900-001-01, Circulomics) and UHMW DNA Aux Kit (NB-900-101-01, Circulomics) according to the manufacturer’s protocol.”*

The reviewer asked us to explain how the protocol we described, which is applicable to adherent cells, could be used for cells growing in suspension.

This protocol is also suitable for suspension cells – we have applied the method to acute myeloid leukemia cell lines (e.g., OCI-AML3). We have updated the text in response to this question to make that point also known. The paper now reads: *“This study focused on two cell lines and four treatments, but the method is usable for any human cell line and is extensible to tumour organoids and suspension cell cultures.”*

Reviewer #2

The reviewer queried whether we could make any observations about fork stalling beyond the association with replication timing. For example, whether fork stalls were detected primarily within genes.

We did not observe any significant differences between fork speed or stall scores within genes, presumably because the probability of a replication-transcription conflict is quite low within the 15-minute labelling window. We did, however, observe an association between fork speed and gene expression in the malaria parasite *Plasmodium falciparum* which indicates that this technology can, in principle, detect such interactions [<https://academic.oup.com/nar/article/51/6/2709/7048502>].

The reviewer made the observation that fork asymmetry is a commonly used metric in DNA fibre analysis to detect fork stalling and queried whether we have sufficient sequencing depth to conduct a similar mode of analysis.

We have added this analysis as the new Supplemental Figure S10. While our treated samples did not show a significant difference (according to a Kolmogorov-Smirnov test) compared to the untreated sample, this approach requires an origin call on a read sufficiently long that neither fork track runs off the read. The Oxford Nanopore reads, while long, are not as long as DNA fibres and this methodology places considerable strain on read length. The result is that we did not get many usable reads for this asymmetry analysis hence, while we fully agree that it was an interesting thing to try, we do not feel that this type of analysis is particularly suited to the method we have developed.

The reviewer wondered whether it was possible to conduct a simple analysis of the number of replication forks detected per unit of input DNA to be used as a proxy for origin density.

This information is already included in Supplemental Table S1.

The reviewer suggested identifying a perturbation to replication from a blinded sample using the replication fork signatures.

This is a very interesting suggestion which we have addressed in Supplemental Figure S7. We partitioned the points in Figure 3(d) according to k-means clustering (k=5 for four treatments plus the untreated case). Supplemental Figure S7 shows the fraction of forks that were partitioned into the correct cluster for the treatment they were given. While performance is generally good across the board, we observe the best performance for PARPi given the distinct phenotype of a fork speed increase. The closest signatures are that of untreated and HU-treated forks; in these conditions, the stall score distribution is similar and the variance of the fork speed distributions is quite wide.

Reviewer #3

The reviewer asked us to discuss whether the PIP-FUCCI system is necessary for the method of whether it is a way to boost throughput. They also asked us to discuss whether alternatives to PIP-FUCCI are possible.

The PIP-FUCCI system is indeed a way to improve the efficiency of the method by improving the number of fork calls per flow cell. It does not affect the results of the method other than by improving the yield of reads with BrdU and EdU labelling by allowing S-phase cells to be enriched by cell sorting. We chose this method because we were wary of cell synchronisation methods that stall replication forks and activate cell cycle checkpoint in a study designed to detect fork stress, but methods such as mimosine arrest or CDK4/6 inhibition and release are, in principle, possible. We added the following text to the manuscript: *“The PIP-FUCCI system was used in this study only to increase the number of fork calls per MinION flow cell; it does not otherwise affect the results. Other cell synchronisation techniques could also be used to enrich S-phase cell, such as mimosine arrest*

or CDK4/6 inhibition and release⁴³. Cell synchronisation techniques may impact fork stress scores depending on the end user's application.”

The reviewer queried whether our method has “up to single-nucleotide resolution” as claimed.

We demonstrated single-nucleotide resolution of BrdU calling in our previous publication and indicated that the drop-off in BrdU occurs at FOB1 binding sites in budding yeast rDNA (<https://bmcgenomics.biomedcentral.com/articles/10.1186/s12864-021-07736-6>). The current model, which detects BrdU and EdU, was trained using the same method. However, quantifying the exact spatial resolution of fork stalling *in vivo* is challenging. Even in the ideal model system of budding yeast rDNA, there are multiple FOB1 binding sites within a single rDNA repeat and it is not necessarily clear which of these sites blocked fork movement on single molecules (<https://doi.org/10.1128/MCB.23.24.9178-9188.2003>). Taking this into account, we feel that “up to single-nucleotide resolution” is the technically accurate characterisation of the method's performance.

The reviewer asked us to explain the rationale for the different treatment drug durations as shown in Figure 2a. They further queried whether it is possible that the different treatment regimens explain the clustering in Figure 3c-d.

The rationale for each treatment condition were to directly comparable with published DNA fibre data and/or perform short treatments to assess the ability of the method to detect rapid changes in replication fork dynamics. Treatment with PARPi Olaparib for 24 hrs is a condition with the ability to accelerate fork speed (Maya-Mendoza, A. et al. Nature 559, 279-284 (2018)). Short treatment with HU treatment and ATRi VE-821 were performed to assess the immediate fork responses following treatments. Short 30 min ATRi treatments were previously performed in multiple papers including (Moiseeva, T et al. Nature Communications (2017)

Drug treatments: HU (2 mM), PARPi Olaparib (10 μ M), Wee1i MK1775 (1 μ M) and ATRi VE-821 (10 μ M).

PARP inhibitor treatments were performed to assess the speed. It is indeed important to emphasise that we expect the clustering in Figure 3c-d to be dependent on the dosage and timing of the agents given. We would, for example, expect to see abrupt fork stalling at higher doses and longer duration of hydroxyurea (than what was used in this study). We had added the following to the manuscript: “While we anticipate variation due to the timing and dosage of treatment, these results show that different chemotherapies can create a characteristic replication stress signature based on the pathway that they target.”

The reviewer asked us to show reads in a representative genomic region with chromosomal coordinates, ideally showing the same region across the different treatment regimens used. They also questioned whether patterns in fork speed and stress differ by chromatin structure, gene density, GC content and other parameters. The actual locations of the regions shown should also be indicated in the figure legends.

While we fully agree with the reviewer that this is a sensible suggestion, coverage of the human genome remains a challenge. Our highest yield runs achieved approximately 4000 fork calls per flow cell, and assuming that these tracks were approximately 20 kb in length, this means we have approximately $(4000 * 20 \text{ kb}) / (3 \text{ Gb}) \approx 3\%$ coverage of the human genome in fork tracks. We have shown, however, that this is possible for smaller genomes such as those of malaria parasites (<https://doi.org/10.1093/nar/gkad093>). We have also shown elsewhere that fork speed decreases in highly AT-rich regions (<https://doi.org/10.1093/nar/gkaf111>).

Different chemotherapies create distinct replication stress signatures: can the authors elaborate on the specific stress form(s) that each drug induces and that lead to the clustering patterns reported in figure 3?

As the stress signature approach is unsupervised, it will be agnostic to any specific stress forms. The only information available to the dimensionality reduction algorithm are the analogue track lengths, stall score, and dynamics of analogue incorporation. It is reasonable to assume that the forms the algorithm identifies tracks with our discussion around the speed and stall scores from each monotherapy: “...compared to untreated cells, HU and WEE1 inhibition resulted in fork slowing and PARP inhibition resulted in a fork speed increase (see Figure 2c), but all three of these chemotherapies resulted in a similar distribution of stall scores to that of untreated cells (Figure 3b). ATR inhibition did not slow forks as much as HU, but treatment with ATR inhibitors resulted in a marked increase in stall score indicative of frequent fork stalling. While HU is generally thought to rapidly stall replication forks, multiple DNA fibre analysis studies have shown slow but continued fork progression during short and prolonged HU treatment^{35, 36}. Our approach builds on these DNA fibre methods by discriminating between fork slowing in cells treated with HU and rapid stalling in cells treated with ATRi.”

Figure 4c-d: the trend is somewhat difficult to see (especially in panel d). A line connecting the bar plots for each time point could make this figure more clear, although this is an option that I would leave to the discretion of the authors.

We have added the results of a Mann-Whitney U test between consecutive time points which should make the trend clearer.

The reviewer asked us to clarify in the discussion whether the method is applicable to non-cancer samples.

We agree that this was not clear and thank the reviewer for pointing this out. We have hence changed the wording to: “...*the method is usable for any human cell line and is extensible to tumour organoids.*”

“a natural example being a new assay for DNA repair”- can the authors elaborate on this? This isn't clear.

We have made clear what we mean by changing the wording to, “... *with an example being incorporation of base analogues during nucleotide excision repair or long-patch base excision repair.*”

“Our software”- as mentioned above, the attributes of the software aren't described in detail in the main text, making it difficult to fully appreciate the novelty of this study.

Model training for DNAscent v3.1.2 (for the R9.4.1 chemistry) is detailed in our publication on *Plasmodium falciparum* replication dynamics (<https://doi.org/10.1093/nar/gkad093>). The novelty in this manuscript is the application to human cells and the introduction of fork stalling and stress signatures, both of which are detailed in the methods section. For DNAscent v4.0.3 (R10.4.1 chemistry) we have detailed in the methods section how this training was carried out and specified the model architecture as Supplemental Figure S8 and Supplemental Table S3.

Reviewer #1

Instead of providing a point-by-point response to my comments, the authors basically cherry picked those they chose to answer to, presenting them in a sometimes incorrectly summarised way, while they simply disregarded other remarks. My concerns regarding the meaning and reliability of the fork “Stall Score”, which is central to the so-called “replication stress signature” of different chemotherapies, were not properly, if at all, taken into account. Yet validating these metrics is all the more important since, whereas nanopore analysis of dual EdU/BrdU pulses presented in this manuscript was clearly new in late 2022, it has now been published twice by the authors (although uniquely in Plasmodium species and not in human cells), with the “Stall Score” and “replication stress signatures” being the real novelties here. Notably, I can’t seem to find the histograms of the EdU/BrdU ratios for each sequencing run that the authors claim to show in Figure S10 (which only present sister fork ratios), not to mention that the fork asymmetry calculation that they used is incorrect. Data were certainly inverted for RPE1 and A2058 cell lines, both in Figure 1 and in the text, and I still wonder what is the actual concentration of hydroxyurea (HU) used in this study (2 mM HU, as written in the Methods section of both the original and revised manuscripts, or 200 μ M HU, as indicated by the authors in their rebuttal letter?). Finally, my questions regarding the accuracy of fork speed measurements by DNAscent still remain. Nonetheless, an important feature of this revised version is the adaptation to Oxford Nanopore Technologies’ R10.4.1 chemistry.

We thank Reviewer #1 for their thoughtful comments and the improvements made to the manuscript as a direct result of their input. We also sincerely apologise that we did not adequately address their previous concerns; this was not our intention. We hope they feel the following text both addresses their concerns and improves the manuscript.

We would like to start by briefly outline the challenges we faced in developing the software, to provide context for our current and previous responses, before addressing each point made by Reviewer #1. The detection of BrdU and EdU is challenging due to the small signal differences between each analogue and thymidine. However, we are confident in our ability to distinguish these signals. The greater challenge has been determining how the start and end of a continuous track of analogue incorporation corresponds to the start and end of the analogue pulses, which is the underlying theme of Reviewer #1’s concerns and comments. We would like to explain how and why our software design decisions were made, as well as the pros and cons of these design decisions, which we have also discussed in the latest draft.

Our approach to developing the DNAscent software has been guided by prior studies using DNA fiber assays. We have set the fork detection to call the start and end of replication tracks at the peaks of EdU and BrdU detection because this is the setup that aligns closest to DNA fiber data. We do not expect the software to always match the fork speeds reported with DNA fiber assays, as these are fundamentally different detection strategies, and it is partly our own design choices that has caused them to be so close.

Important considerations for the differences between the two detection methods are:

- **Thresholds for detection:** It is unclear how well the antibodies can detect low levels of CldU and IdU incorporation. DNAscent can detect low levels of analogue incorporation and mixed incorporation when the incorporation rate of both analogues are low. We have noticed

that the BrdU labelling is capable of rapidly saturating the prior EdU signal, although the EdU signal frequently returns as BrdU drops which suggests a different half-life for each analogue.

- **DNA Fiber Challenges:** There are many examples of experimental variability, staining challenges (overlapping signals), variability in DNA stretching, inconsistent use of ssDNA detection to control for total fiber length, and differences reported with spreading versus combing, etc.
- **DNAscnt Challenges:** It is far more sensitive at analogue detection, which can provide long tails of incorporation that are not considered in fork speed calculation but are used in fork stall scores. It is unclear what causes stall vs. pause, and how long they remain stopped (fibres restart challenge = stalled, changes in speed = stalled or asymmetric). Shorter pulses (5+10 mins vs 20 +20 mins) are performed to capture entire fork tracts on a read N50 of 60-110 Kb. This translates to forks being monitored for less time or less probability of capturing a change in fork behaviour during the labelling.

Here we provide a point by point response to Reviewer #1 comments.

Major concerns:

1) Whereas I previously expressed strong concerns regarding the Stall Score, stating that « « the Stall Score must be fully validated before being used as a trustable marker of fork stalling », there is no additional validation in the revised version of the manuscript, since the EdU/BrdU ratios I asked for are nowhere to be found in Figure S10 despite the authors's claim. Actually, the sister fork analysis did not show any significant difference between the treated and untreated samples, in contrast to the authors' conclusions for ATR inhibition using the Stall Score, but the number of values (which the authors must provide in Figure S10) was apparently low and possibly insufficient. No such limitations should apply to the EdU/BrdU ratios, that I once again demand to be computed.

Looking back at Reviewer #1's comments, it is now clear that we misunderstood the query for which we sincerely apologise. We are happy to provide the EdU/BrdU ratios but, because of how the software works, the values are different to the ratios observed with DNA fibres and do not provide a straightforward validation of the stall score, as intended. It is unclear what exactly these ratios represent as they vary considerably across the individual repeats and the treatment conditions. Therefore, we haven't included these ratios in the revised manuscript. To acknowledge the reviewers valid criticism, we have added a section on the technical considerations and limitations to the discussion that mentions the points above and the lack of validation of the stall score.

The following plot shows the EdU/BrdU ratios for each independent experiment for the different treatment conditions. For each fork call, the length of the BrdU track has been divided by the length of the EdU track (both in bp). It is noteworthy that these ratios are quite a bit higher than would be expected based on the 5-min EdU pulse and the 10-min BrdU pulse. We understand reviewer's desire to query this based on what was observed in Figure 1f and Figure 2b.

To understand how these high ratios arise, we can consider, without loss of generality, the track of a rightward-moving fork. From left to right, there are four points of interest: the (1) start and (2) end of the EdU track and the (3) start of the BrdU track and (4) point where the BrdU incorporation begins to trail off. The fork track length is calculated by calculating (4)-(1), and we calculate speed by dividing this track length by the total pulse time of 15 minutes. We are confident in the locations of (1) and (4) given the ability of the software to provide a match to fork speed data calculated via DNA fibre. However, the locations of (2) and (3) can have an important impact on the ratios that the reviewer requested, but we expect that the software will provide worse estimates of these based on how it works.

To finalise the location of the start and end of the EdU track, the software takes the middle third of the EdU track and it computes the average incorporation there. It then trims back from the start of the track and the end of the track (where incorporation is lower) until it meets a part of the segment with this average incorporation. The trimming then stops and this is given as the final track. The software does the same for the BrdU track. We did this for two reasons:

- It allows us to calculate where BrdU incorporation begins to trail off so that we know where to place (4).
- It makes sure the segmentation of the EdU-to-BrdU transition is clean, which is extremely important. In the example of the rightward-moving fork, from left to right, suppose the software correctly calls the transition from EdU to BrdU, but then the software incorrectly flips back to EdU for a short segment before it goes back to BrdU. This would represent a serious error: There is now an origin and a termination site in what should have been one rightward-moving fork track. A fork track where this error could have occurred is in the ATRI track of Figure 2b. We can see that the software has avoided this error and called the EdU and BrdU tracks as intended.

The single-molecule nature of the method means that we are in a challenging engineering situation where calling even 95% of molecules correctly is poor: It means that 1 in 20 fork tracks are wrong. Such situations require trade-offs where more serious errors can be prevented at the expense of minor errors; it is not possible for algorithms to be perfect everywhere, on everything, all the time. Hence, we designed this software under the assumption that the location of the EdU-to-BrdU transition relatively unimportant compared to approximating the fork speed correctly and avoiding a far more serious error such as confusing a rightward-moving fork with an origin and termination.

In addition, a further opportunity for higher-than-expected BrdU/EdU ratios comes from the length of the analogue pulses. These ratios come from DNA fibre where the molecules are longer and, hence, analogue pulse lengths can also be longer. The result is ratios that are robust to minor inaccuracies in track length. While the Oxford Nanopore platform provides long reads, they are still considerably shorter than what would be used in DNA fibre. This motivated our choice of the relatively short pulse lengths of a 5-min EdU pulse and a 10-min BrdU pulse. However, particularly for the EdU pulse, this means that minor inaccuracies in the estimation of track length (± 1 kb would be well within the margin of error) can yield very high ratios. Again turning to Figure 2b, we see a clear example of this for the WEE1i track. The transition from EdU to BrdU is mixed, with a ~ 4 kb region between the EdU and BrdU tracks showing moderate levels of both analogues. It is ambiguous exactly where the transition is so, as intended, the software calls the tracks conservatively. This results in an EdU track that may be too short with the potential to produce a high BrdU/EdU ratio.

In Figure S10, the quantity $|(left\ fork\ track\ length)/(right\ fork\ track\ length) - 1|$ is not accurate to compute fork asymmetry. For instance, if a left fork track length is twice that of the right one, the asymmetry score will be $abs(2-1)=1$, but if the right fork track length is twice that of the left one, the asymmetry score will be $abs(0.5-1)=0.5$, whereas it should be 1 in both cases. The ratio between the longest and shortest fork track lengths must be used instead.

We have updated the results in Figure S10 using the metric the reviewer requested.

I wonder why the number of fork calls for the Stall Score is different from the first version of the manuscript in Figures 3b, 4a, S5 and S6a, b.

This is a good question and observation. When we measure replication fork speed, we exclude any replication fork whose track is directly adjacent to its sister fork in an origin, or whose track is directly adjacent to an opposing fork in a termination. These fork tracks will have been shortened due to origin firing or termination during the pulse and can lead to a biased measure of fork speed if not excluded. While this exclusion is important for fork speed, it is not strictly necessary for stall score. For a replication fork whose track is directly adjacent to its sister fork in an origin, the point where BrdU incorporation begins to trail off can still be observed provided it is not too close to the end of the read. However, for consistency, we still felt that it was marginally preferable to consider largely the same group of forks for both fork speed and stall score. The very subtle difference in the N numbers that still exists between the two is due to a minor difference in how read ends are handled. While this was certainly good to query, the shape of the distribution in each case is nearly identical.

As previously written, the authors must sustain their claims regarding similar or dissimilar Stall Score distributions with an appropriate statistical analysis.

Given that we have two replicates for each treatment of A2058, we built hierarchical Bayesian models of both fork speed and stall score for each treatment to show whether the treatment effect exceeds variation between the two untreated replicates. Of the statistical options available, we feel that this method will present the most nuanced level of detail to readers should they wish to see it. We have also done the same for HCT116 to compare WT to $CDK2^{AF/AF}$. The full results are displayed in new Supplemental Figures S5 (fork speed for A2058), S7 (stall score for A2058), and S11 (speed and stall score for HCT116 WT vs $CDK2^{AF/AF}$). We have defined a Region of Practical Equivalence (ROPE), i.e., an effect size sufficiently close to zero that it cannot be meaningfully distinguished. We set this value at ± 0.05 for stall score and ± 0.1 kb/min for replication fork speed. These results support our original claim that all treatments produce meaningful changes in fork speed relative to the untreated control, and that only ATR inhibition produces a meaningful change to stall score.

2) In my earlier comments, I did not only « queried (the authors') treatment protocol with hydroxyurea, particularly when the treatment was given », but pointed out that the weak fork slowdown measured after a 2 mM HU treatment was in sharp contrast with the expected drastic decrease in fork speed in the presence of such a high drug concentration, as recurrently reported in the literature. Now that the authors have explained, in Figure S3, how fork speed with HU was computed, I must stress again that they cannot take into account both the EdU (« rest of the fork ») and BrdU tracks to calculate fork velocity with HU, since they are mixing two different experimental

conditions. Only the BrdU track, ie, the portion of the signal obtained in the presence of HU, must be used. The way the « replication stress signature » was calculated for HU must also be modified accordingly.

The original intent of our HU treatment protocol was to serve as an internal control at the fork level in order to ensure that there is a significant shortening of the BrdU track while the length of the EdU track remains largely unchanged. This protocol therefore provides important validation, and we feel that Figure S3 supports the fact that the software works as intended. However, we agree with the reviewer's point that this creates the challenge of how to appropriately compare the HU-treated forks to the other datasets presented in this study. As discussed in response to the reviewer's first point in this document, we are not confident in using the BrdU track alone to reliably measure fork speed across all samples used in the study due to a possible lack of precision at detecting the EdU-BrdU boundary. Therefore, using the BrdU track alone across all samples would not be a viable way forward. Moreover, as the replication signatures used in Figure 3 are new to this study, we are hesitant to fundamentally change how this works for a sample which, as shown by Supplemental Figure S8, already clusters relatively well – despite the fact that the way we have calculated puts it at a disadvantage by making it harder to distinguish from the untreated sample.

To find a harmonious solution to this problem, we now present two versions of all analyses involving HU-treated A2058 cells. In particular:

- Figure S5 (statistics on the treatment effect) shows plots labelled “HU (EdU and BrdU tracks)” and “HU (BrdU track only)”. The former uses fork length as defined in Figure 2b for both the untreated and HU-treated sample, while the latter uses only the length of the BrdU track for both the untreated and HU-treated sample.
- Figure S9 repeats the signature analysis shown in Figure 3c but uses the length of the BrdU track as the only metric of fork speed.

Given the BrdU track length difference observed between untreated and HU-treated conditions in Figure S3, speed is apparently reduced by a factor of 3 with HU. This is somewhat compatible with a treatment with 200 μ M HU, the concentration indicated by the authors in the rebuttal letter. However, forks would still be moving abnormally fast if cells had been exposed to 2 mM HU, the dose indicated in the Methods section of both the original and revised manuscripts. So I wonder: what is the actual concentration of HU used in this study?

This was a mistake in the previous response-to-reviewers document for which we apologise. The sentence should have read, “As shown by Figure 2a, we treated cells first with EdU for 5 minutes and then treated cells with 2 mM of HU together with BrdU.” The concentrations, as stated in the manuscript, are correct.

While we appreciate that a fork travelling 4-5 kb in 2 mM of HU may initially be surprising, this again is likely due to a difference between our method and DNA fibre. A distance of 4-5 kb is not far above the spatial resolution of DNA fibre analysis. Moreover, our 10-minute BrdU+HU treatment is considerably shorter than what would be used in DNA fibre analysis (the reviewer cited Somyajit et al. 2017 and Koundrioukoff et al. 2013 in their previous response who used a 30-minute and 60-minute HU treatment, respectively). It may be the case that there is a short (but non-trivial on the timescale of 10 minutes) period when HU washes in, progressively slows down the fork, and

ultimately causes it to stop which would harmonise what we observe and measure with what the reviewer expects. However, this is still fundamentally different behaviour than a fork stopping suddenly which is what our stall score was designed to measure.

3) Commendably, in this revised version, the authors assessed fork speed in an additional cell line, namely RPE1, finding a fork speed consistent with previous studies. However, I am puzzled by the difference between DNA fiber spreading and DNAscent results for A2058 cells, which gave fork speed estimates of 1-1.13 kb/min and 1.46-1.52 kb/min, respectively. In other words, forks appear to move 30-50% faster in A2058 cells when their speed is measured by DNAscent instead of DNA fiber spreading, which is far from insignificant. Although none of these two approaches can be considered as giving the « ground truth », the authors uniquely blame DNA fiber analysis for these discrepancies without questioning their own method, which I think they should.

Previously, since I was not necessarily convinced by DNAscent's segmentation of the reads into EdU- and BrdU-positive regions, I asked the authors to « report how the whole fork calling procedure was validated » since « such details are critically important to evaluate forkSense accuracy ». According to the data presented in Figure S3, I feel that my doubts were well-founded. Indeed, it appears that, in untreated cells, BrdU tracks are 3 times longer than EdU tracks (« rest of fork »), whereas a ratio of 2 is expected given that BrdU labelling time is only twice that of EdU. Such an overestimation of BrdU track length alone could almost entirely explain the difference in speed with the DNA fiber analysis. A possibly biased evaluation of BrdU track length does not really come as a surprise in view of the exemplary reads shown in Figs. 1 and 2, since positioning the end of the fork track « where BrdU incorporation starts to decrease at the start of the thymidine chase” do seem more complicated than positioning the start of the EdU track or the EdU/BrdU transition. Because the Stall Score is also based on the analysis of the BrdU level within BrdU tracks, this issue must be thoroughly addressed.

We appreciate the reviewer's concern over Figure S3. As this is really part of the larger discussion about track length ratios of EdU-to-BrdU and how the analogue tracks are segmented, we refer back to the previous discussion in this document where we aimed to provide a thorough description of this issue.

4) Regarding the duration of the wash, I previously anticipated it to be negligible compared to the 15 minutes of EdU/BrdU labelling time. In fact, washes last for 3-4 minutes according to the authors, and I therefore wonder why fork speed is currently computed by dividing the length of the EdU/BrdU labelled region by 15 minutes and not 18-19 minutes, which is the actual time separating the start of the EdU pulse and the start of the thymidine chase. This likely results in an overestimation of fork speed although, obviously, modifying the time interval will only impact « absolute » speed values and does not prevent comparisons between untreated and treated conditions.

This is certainly a fair point and something we thought about quite a bit. One reason we calculated it this way is for consistency with DNA fibre: Wash time is not typically included in the fork speed calculation from DNA fibre and the duration of the wash for DNAscent and DNA fibre is similar. We appreciate that the analogue pulse durations are shorter for DNAscent than they are for DNA fibre, and hence the wash duration may make more of a difference for our method. However, as detailed above, the dynamics of analogue incorporation into nascent DNA is complex and we do not generally observe a “gap” in analogue incorporation resulting from the wash. We therefore feel that

fork speed measurements from both DNAscent and DNA fibre should be treated as approximate, and while speed measurements from the two methods should be generally similar as we have shown, subtle differences in how fork speed is calculated means that they will not be exactly comparable. As the reviewer says, this only pertains to the absolute fork speed. All of the treated samples in his manuscript are accompanied by an untreated sample.

5) The authors are requested to respond to any comments from my first review that they have not yet addressed.

We apologise for any oversight that may have occurred. We believe that the main points were to do with fork speed measurements under HU and the EdU/BrdU ratios which we have now addressed above, but responses to additional points are below.

The “Fork calling with DNAscent” paragraph should also extensively describe how the start and end of replication forks were determined, specifying how EdU- and BrdU-positive regions within a read were segmented by forkSense.

The following text was added to the methods section: *“The DNAscent forkSense subprogram segmented each read into EdU- and BrdU-positive regions. To finalise the location of the start and end of the EdU track, DNAscent computes the average incorporation in the middle third of the track and trims back from the start of the track and the end of the track until it meets a part of the track with this average incorporation. The trimming then stops and this is given as the final track. DNAscent then does the same for the BrdU track. EdU and BrdU tracks are then matched into replication forks such that, for a rightward-moving fork, the replication fork track begins at the start of the EdU track and ends at the end of the BrdU track which is where BrdU incorporation starts to decrease at the start of the thymidine chase.”*

As stated above, I truly wonder how the fork stalling scoring procedure was validated apart from comparing with previous biological results. How was the non-linear scaling “approved”, for instance?

As shown previously (<https://www.nature.com/articles/s41592-019-0394-y>) BrdU incorporation drops off suddenly when forks are blocked by replication fork barriers in budding yeast rDNA. The raw score that we refer to as R in the methods section measures the size of this drop-off. The application of nonlinear scaling, which converts R into a stall score, was designed to make this process more conservative in order to prevent false positives. The following plot shows the nonlinear scaling applied by DNAscent v3.1.2 where we see that quite high levels of R are required to produce a high stall score.

The whole manuscript would in fact greatly benefit from a systematic comparison between fork speed/stalling estimated either by DNAscent or by standard DNA fiber analysis for all cell lines and culture conditions.

We are unable to action this due to a lack of DNA fibre data on the A2058 cell line. We have performed DNA fibre for the untreated case, but we are not in a position to do so for all of the conditions presented in this manuscript.

Please specify the duration of the wash between the EdU and BrdU pulses, which is not taken into account in the total time used to calculate fork speed.

While we responded to this point in our previous document, we appreciate that it was not added to the manuscript at the time. This has now been fixed and the effect of the wash duration on fork speed calculation has been commented on above.

The replication timing may be different in HCT116 CDK2AF/AF mutant compared to WT cells, especially since this mutant displays aberrant replication dynamics (Hughes et al, 2013). The authors either need to provide evidence of similar timing in both cell lines or tune down their conclusions.

To make this clear, we have updated the wording of the following sentence:
 “This relationship vanished in the CDK2^{AF/AF} mutant, as these cells maintained a constant slow fork speed and high stall score across both early- and late-replicating regions of the genome.”

to new wording that makes this point clear:
 “This relationship vanished in the CDK2^{AF/AF} mutant, as these cells maintained a constant slow fork speed and high stall score across regions of the genome that replicated both early and late in wild-type cells.”

How is fork speed estimated by DNAscent affected by read length?

Fork speed estimates by DNAscent will be affected by read length in a similar manner to how fork speed estimated by DNA fibre is affected by fibre length. While we were pleased to be able to produce very long (N50 ≥ 74 kb) reads relative to what would be typical on the

ONT platform, these reads are still quite a bit shorter than fibres. We compensated for this by making our pulse durations correspondingly shorter. Importantly, we have the ability to easily determine where the analogue tracks are in relation to the end of the molecule without the need to stain for DNA. We therefore exclude molecules from fork speed calculations which are within 3 kb of the end of the molecule as we have described in the methods section.

Other points to be addressed:

- Update reference 23.

This has been fixed.

- « as well as the malaria parasites Plasmodium falciparum²³ and Plasmodium knowlesi²⁴ »: please add a reference to the article « The genetic landscape of origins of replication in P. falciparum » by Casilda Muñoz Castellano et al, Nucleic Acids Research, 2024.

This reference has been added.

- « replicates with fork speeds of ~1.5 kb/min in RPE1 cells »: please replace 1.5 with 1.4 since RPE1 and A2058 cell lines have certainly been inverted in Fig. 1e.

This has been fixed, and we appreciate the reviewer marking us aware of the issue with Figure 1e.

- « a difference of approximately 0.2-0.3 kb/min »: please replace 0.2-0.3 with 0.3-0.5 since RPE1 and A2058 cell lines have certainly been inverted in Fig. 1e.

This has been fixed.

- « This study focused on two cell lines »: please replace two with three.

This has been fixed.

- Figure 1e: A205 and RPE1 cell lines have certainly been inverted; please correct.

This has been fixed.

- Figure 5b caption: Distribution of replication fork speeds AND stall scores.

This has been fixed.

- Figure S3a caption: remove « was measured ».

This has been fixed.

- *Figure S6 caption: Biological replicate OF the experiments shown in Figure 4.*

This has been fixed.

- *Figure S8 caption: Supplemental Table S3 shows the values OF the number of filters.*

This has been fixed.

- *Figure 3c: please show the centroids for each biological replicate separately. The untreated condition is hardly clustering; could the authors explain why? (see also my related comment regarding Figure S7).*

- *Figure S7: the clustering of the untreated sample is not very convincing, since it is called more like an HU-treated sample. This is possibly related to the issue mentioned above concerning the way the HU sample was handled, with part of its signal corresponding to untreated conditions. Also, to really show « how good the clustering is », as the authors state, it seems important to perform this analysis independently for both replicates of each category.*

We fully appreciate the sentiment of what the reviewer is suggesting in the above two comments and agree that validation of the clustering is important. However, if we understand correctly, there are some challenges in carrying out exactly what the reviewer has asked us to do.

In the first point, we understand this to mean that they would like us to compute a centroid for the replicate of each treatment rather than the centroid of each pooled treatment. However, it is important to note that Figure 3c and 3d do not show the centroids of a treatment (pooled or otherwise). The signatures were mapped down to two dimensions using UMAP and we then applied K-means clustering with $k=5$. The centroids shown are the centroids of the five clusters identified by the K-means algorithm. Both of these steps were blind to which treatment and which replicate a particular point corresponded to. The validation lies in the fact that K-means clustering points out centroids that correspond to the clusters we see by eye.

In the second point, we understand this to mean that the reviewer would like us to perform this procedure twice, using only one treatment replicate for each. However, comparing the results of two different UMAP embeddings in this way would not be mathematically sound because the scale and orientation of each embedding is independent. While, in principle, a Procrustes superimposition or similar may be possible, we feel this would be excessively convoluted and nonstandard, hence we would be reluctant to believe any conclusion that came from it.

Ultimately, while these are both interesting ideas, our analysis is the standard approach (see, for example, *Introduction to Information Retrieval* by Manning, Raghavan, and Schütze) and is more quantitative than either of the approaches proposed above. Therefore, we feel that this analysis is stronger in its current form. However, we appreciate the reviewer's previous suggestion to include the untreated condition which has improved the manuscript.

The reviewer has made a good observation that, compared to the other treatments, the clustering of the untreated sample is not as strong. This comes from the shape of the fork speed and stall score distributions. As shown in Supplemental Figure S7, only ATRi shows a strong change in the

distribution of stall score. For fork speed, HU is the slowest with a low variance. While the fork speed distributions for ATRi and WEE1i are similar, they can be distinguished from one another by stall score. Finally, PARPi is an outlier with a faster fork speed. For these reasons, it is relatively straightforward to distinguish these four treatments from one another. However, when we consider the untreated sample, the variance in fork speed is comparatively high and it has a similar stall score distribution to ATRi, HU, WEE1i, and PARPi. While the other features in the signature add some distinguishing power, this lack of any large distinguishing feature is enough to make the untreated cluster quite diffuse.

Reviewer #1

I thank the authors for their long and honest answer regarding the limitations of this study. I still have my doubts regarding the stall score, but research necessarily involves some uncertainty. However, only one discussion paragraph has been added to the main text regarding these limitations, and I think that readers would benefit from some of the information provided by the authors in their rebuttal letter being directly incorporated into the manuscript (I am aware that peer review files are public, but they are inevitably less consulted than the article itself). Several important additional analyses have also been performed in the revised version, which is commendable.

We would like to thank Review #1 again for the time and care they have spent on our manuscript. They have made many excellent points over the course of the revision process, and we feel that the manuscript is now markedly better because of their feedback.

One of those excellent points was to do with the EdU-to-BrdU track length ratios, so we have added the following additional paragraph to the discussion which both highlights this point and guides users on our suggested way to use DNAscent based on the software design choices we have made:

“When engineering DNAscent, our design choices were made to prioritise the accuracy of replication fork speed and stall score. This necessarily required trade-offs in other areas, particularly on the location of the boundary between the EdU and BrdU regions of a fork track. We often observe noise at the EdU-to-BrdU transition whereby EdU incorporation is elevated within the BrdU region, as well as ambiguity over where this boundary should be placed. These two phenomena are visible, respectively, in the representative ATRi and WEE1i fork tracks of Figure 2b. Our design choices were made to minimise risk of the major segmentation error that would occur if, for example, a short segment of elevated EdU incorporation within a BrdU region was mistakenly called as an origin rather than ignored as noise. By making the calling of EdU and BrdU regions conservative to avoid these errors, DNAscent tends to leave a gap between the EdU and BrdU regions of a fork track. This can sometimes result in a ratio of BrdU-to-EdU track lengths that do not match the expected 2:1 ratio given the 5-minute EdU pulse and 10-minute BrdU pulse. Therefore, while we have added HU with the BrdU pulse in this study to serve as an internal control to verify shortening of the BrdU track relative to the EdU track (Figure S3), we would not generally recommend this experimental setup to test the effectiveness of an inhibitor on replication fork dynamics. We instead suggest adding the inhibitor with or before the first analogue pulse.”

Minor comments:

1. “This relationship vanished in the CDK2AF/AF mutant, as these cells maintained a constant slow fork speed and high stall score across regions of the genome that replicated both early and late in wild-type cells”: even if the wording is now correct, I still wonder what this really means given that replication timing may be different in CDK2AF/AF than in WT cells.

We agree that this is a good point and that it is possible that the CDK2^{AF/AF} mutant may have a different replication timing profile compared to the wild type. As the replication timing profile of human cells is largely conserved, even amongst cancer cells, it would be surprising if the effect was

significant enough to produce the total loss of correlation between location, speed, and stress observed in Figure 4. Nevertheless, we agree that complementing this observation with Repli-seq would be a useful direction for future work. We have included the following sentence to make this point: *“Our study has not accounted for any potential changes in replication timing in the HCT116 CDK2AF/AF and could have been improved by generating equivalent high resolution Repli-Seq timing reference for this cell line.”*

2. The sentence “While we observed a higher variability... that exceeded this variability (Figure S11)” lacks clarity and may be rephrased.

We have now changed this sentence to read:

“While we observed a higher variability between repeats than in A2058 (wild-type median fork speed of 1.7 k/min with stall score median 0.3; CDK2^{AF/AF} median fork speed of 1.26 kb/min with stall score median 0.39; Figure S10) the CDK2^{AF/AF} cells showed a slower fork speed and higher stall score than the wild-type within each repeat (Figure 4a-b). Moreover, there was moderate support for a treatment effect of the CDK2^{AF/AF} mutation (Figure S11).”

3. “Stress score” is sometimes used in this manuscript instead of “stall score”; this may be homogenised.

We appreciate the reviewer pointing this out and we have now changed all terms to “stall score” for uniformity and clarity.